# Posterior Behavioral Cloning: Pretraining BC Policies for Efficient RL Finetuning

## Abstract

Standard practice across domains from robotics to language is to first pretrain a policy on a large-scale demonstration dataset, and then finetune this policy, typically with reinforcement learning (RL), in order to improve performance on deployment domains. This finetuning step has proved critical in achieving human or super-human performance, yet while much attention has been given to developing more effective finetuning algorithms, little attention has been given to ensuring the pretrained policy is an effective initialization for RL finetuning. In this work we seek to understand how the pretrained policy affects finetuning performance, and how to pretrain policies in order to ensure they are effective initializations for finetuning. We first show theoretically that, by training a policy to clone the demonstrator's *posterior* distribution given the demonstration dataset—rather than simply the demonstrations themselves—we can obtain a policy that ensures coverage over the demonstrator's actions—a minimal condition for effective finetuning—without hurting the performance of the pretrained policy. Furthermore, we show that standard behavioral cloning (BC) pretraining fails to achieve this without significant tradeoffs in terms of sampling costs. Motivated by this, we then show that this approach is practically implementable with modern generative policies in robotic control domains, in particular diffusion policies, and leads to significantly improved finetuning performance on realistic robotic control benchmarks, as compared to standard behavioral cloning.

## 1 Introduction

Across domains—from language, to vision, to robotics—a common paradigm has emerged for training highly effective "policies": collect a large set of demonstrations, "pretrain" a policy via behavioral cloning (BC) to mimic these demonstrations, then "finetune" the pretrained policy on a deployment domain of interest. While pretraining can endow the policy with generally useful abilities, the finetuning step has proved critical in obtaining effective performance, enabling human value alignment and reasoning capabilities in language domains (Ouyang et al., 2022; Bai et al., 2022a; Team et al., 2025; Guo et al., 2025a), and improving task solving precision and generalization to unseen tasks in robotic domains (Nakamoto et al., 2024; Chen et al., 2025; Kim et al., 2025; Wagenmaker et al., 2025). In particular, reinforcement learning (RL)-based finetuning—where the pretrained policy is deployed in a setting of interest and its behavior updated based on the outcomes of these online rollouts—is especially crucial in improving the performance of a pretrained policy.

Critical to achieving successful RL-based finetuning performance in many domains—particularly in settings when policy deployment is costly and time-consuming, such as robotic control—is sample efficiency; effectively modifying the behavior of the pretrained model using as few deployment rollouts as possible. While significant attention has been given to developing more efficient finetuning algorithms, this ignores a primary ingredient in the RL finetuning process: the pretrained policy itself. Though generally accepted that a stronger pretrained policy is a better initialization for finetuning (Guo et al., 2025a; Yue et al., 2025), it is not well understood how pretraining impacts finetuning performance beyond this, and how we might pretrain policies to enable more efficient RL finetuning.

In this work we seek to understand the role of the pretrained policy in RL finetuning, and how we might pretrain policies that (a) enable efficient RL finetuning, and (b) before finetuning, perform no worse than the standard BC policy. We propose a novel pretraining approach—*posterior behavioral*

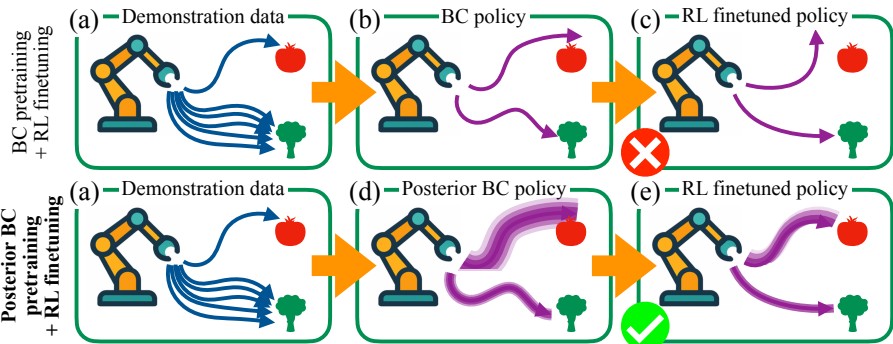

Figure 1: We consider the setting where we are given demonstration data for some tasks of interest, (a). (b) Standard BC pretraining fits the behaviors in the demonstrations, leading to effective performance in regions with high demonstration data density, yet poor performance in regions with low data density. (c) This leads to ineffective RL finetuning, since rollouts from the BC policy provide little meaningful reward signal in such low data density regions, which is typically necessary to enable effective improvement. (d) In contrast, we propose *posterior behavioral cloning*, which instead of directly mimicking the demonstrations, trains a generative policy to fit the *posterior distribution* of the demonstrator's actions. This endows the pretrained policy with a wider distribution of actions in regions of low demonstrator data density, while in regions of high data density it reduces to approximately the standard BC policy. (e) This wider action distribution allows for collection of diverse observations with more informative reward signal, allowing for more effective RL finetuning.

*cloning*—which, rather than fitting the empirical distribution of demonstrations as standard BC does, instead fits the *posterior* distribution over the demonstrator's behavior. This enables the pretrained policy to take into account its potential uncertainty about the demonstrator's behavior, and adjust the entropy of its action distribution based on this uncertainty. In states where it is uncertain about the demonstrator's actions, posterior BC samples from a high-entropy distribution, allowing for a more diverse set of actions that may enable further policy improvement, while in states where it is certain about the demonstrator's actions, it samples from a low-entropy distribution, simply mimicking what it knows to be the (correct) demonstrator behavior (see Figure 1).

Theoretically, we show that posterior BC leads to provable improvements over standard BC in terms of the potential for downstream RL performance. In particular, we focus on the ability of the pretrained policy to cover the demonstrator policy's actions—whether it samples all actions the demonstrator policy might sample—which, for finetuning approaches that rely on rolling out the pretrained policy, is a prerequisite for ensuring finetuning can even match the performance of the demonstrator. We show that standard BC can provably fail to cover the demonstrator's distribution, while posterior BC *does* cover the demonstrator's distribution, incurs no suboptimality in the performance of the pretrained policy as compared to the standard BC policy, and achieves a near-optimal sampling cost out of all policy estimators which have suboptimality no more than the BC policy's.

Inspired by this, we develop a practical approach to approximating the posterior of the demonstrator in continuous action domains, and instantiate posterior BC with modern generative models—diffusion models—on robotic control tasks. We demonstrate experimentally that posterior BC pretraining can lead to significant performance gains in terms of the efficiency and effectiveness of RL finetuning, as compared to running RL finetuning on a policy pretrained with standard BC, and achieves these gains without decreasing the performance of the pretrained policy itself. We show that this holds for a variety of finetuning algorithms—both policy-gradient-style algorithms, and algorithms which explicitly refine or filter the distribution of the pretrained policy—enabling effective finetuning performance across a variety of challenging robotic tasks.

## 2 RELATED WORK

**BC pretraining.** BC training of expressive generative models—where the model is trained to predict the next "action" of the demonstrator—forms the backbone of pretraining for LLMs (Radford et al., 2018) and robotic control policies (Bojarski, 2016; Zhang et al., 2018; Rahmatizadeh et al., 2018; Stepputtis et al., 2020; Shafiullah et al., 2022; Gu et al., 2023; Team et al., 2024; Zhao et al.,

2024; Black et al., 2024; Kim et al., 2024). We focus in particular on policies parameterized as diffusion models (Sohl-Dickstein et al., 2015; Ho et al., 2020; Song et al., 2020), which have seen much attention in the robotics community (Chi et al., 2023; Ankile et al., 2024a; Zhao et al., 2024; Ze et al., 2024; Sridhar et al., 2024; Dasari et al., 2024; Team et al., 2024; Black et al., 2024; Bjorck et al., 2025). These works, however, simply pretrain with standard BC, and do not consider how the pretraining may affect RL finetuning performance.

**Other approaches for pretraining from demonstrations.** While our primary focus is on behavioral cloning (as noted, the workhorse of most modern applications) other approaches to pretraining from demonstrations exist. BC is only one possible instantiation of *imitation learning*; other approaches to imitation learning include inverse RL (Ng et al., 2000; Abbeel & Ng, 2004; Ziebart et al., 2008), methods that aim to learn a policy matching the state distribution of the demonstrator, such as adversarial imitation learning (Ho & Ermon, 2016; Kostrikov et al., 2018; Fu et al., 2017; Kostrikov et al., 2019; Ni et al., 2021; Garg et al., 2021; Xu et al., 2022; Li et al., 2023b; Yue et al., 2024), and robust imitation learning (Chae et al., 2022; Desai et al., 2020; Tangkaratt et al., 2020; Wang et al., 2021; Giammarino et al., 2025). The majority of these works, however, either assume access to additional data sources (e.g. suboptimal trajectories), or require online environment access and are therefore not truly offline pretraining approaches, the focus of this work. Furthermore, none of these works explicitly consider the role of pretraining in enabling efficient RL finetuning.

Meta-learning directly aims learn an initialization that can be quickly adapted to a new task. While instantiations of meta-learning for imitation learning exist (Duan et al., 2017; Finn et al., 2017b; James et al., 2018; Dasari & Gupta, 2021; Gao et al., 2023), our setting differs fundamentally from the meta-imitation learning setting. Meta-imitation learning assumes access to demonstration data from *more than one task*, and attempts to learn an initialization that will allow for quickly adapting to demonstrations from a *new* task. In contrast, we primarily consider learning on a *single* task (though our approach does extend to multi-task learning), and aim to find an initialization that allows for improvement on the *same* task, while preserving pretrained performance on this task. Furthermore, rather than learning from new *demonstrations*, as meta-imitation learning does, we aim to learn from (potentially suboptimal) data collected online and that is labeled with rewards.

**RL finetuning of pretrained policies.** RL finetuning of pretrained policies is a critical step in both language and robotic domains. In language domains, RL finetuning has proved crucial in aligning LLMs to human values (Ziegler et al., 2019; Ouyang et al., 2022; Bai et al., 2022a; Ramamurthy et al., 2022; Touvron et al., 2023), and enabling reasoning abilities (Shao et al., 2024; Team et al., 2025; Guo et al., 2025a). A host of finetuning algorithms have been developed, both online (Bai et al., 2022b; Bakker et al., 2022; Dumoulin et al., 2023; Lee et al., 2023; Munos et al., 2023; Swamy et al., 2024; Chakraborty et al., 2024; Chang et al., 2024) and offline (Rafailov et al., 2023; Azar et al., 2024; Rosset et al., 2024; Tang et al., 2024; Yin et al., 2024). In robotic and control domains, RL finetuning methods include directly modifying the weights of the base pretrained policy (Zhang et al., 2024; Xu et al., 2024; Mark et al., 2024; Ren et al., 2024; Hu et al., 2025; Guo et al., 2025b; Lu et al., 2025; Chen et al., 2025; Liu et al., 2025), Best-of-$N$ sampling-style approaches that filter the output of the pretrained policy with a learned value function (Chen et al., 2022; Hansen-Estruch et al., 2023; He et al., 2024; Nakamoto et al., 2024; Dong et al., 2025b), "steering" the pretrained policy by altering its sampling process (Wagenmaker et al., 2025), and learning smaller residual policies to augment the pretrained policy's actions (Ankile et al., 2024b; Yuan et al., 2024; Jülg et al., 2025; Dong et al., 2025a). Our work is tangential to this line of work: rather than improving the finetuning algorithm, we aim to ensure the pretrained policy is amenable to RL finetuning.

**Posterior sampling and exploration.** Our proposed approach relies on modeling the posterior distribution of the demonstrator's actions. While this is, to the best of our knowledge, the first example of applying posterior sampling to BC, posterior methods have a long history in RL, going back to the work of Thompson (1933). This works spans applied (Osband et al., 2016a;b; 2018; Zintgraf et al., 2019) and theoretical (Agrawal & Goyal, 2012; Russo & Van Roy, 2014; Russo et al., 2018; Janz et al., 2024; Kveton et al., 2020; Russo, 2019) settings. More generally, our approach can be seen as enabling BC-trained policies to *explore* more effectively. Exploration is a well-studied problem in the RL community (Stadie et al., 2015; Bellemare et al., 2016; Burda et al., 2018; Choi et al., 2018; Ecoffet et al., 2019; Shyam et al., 2019; Lee et al., 2021; Henaff et al., 2022), with several works considering learning exploration strategies from offline data (Hu et al., 2023; Li et al., 2023a; Wilcoxson et al., 2024; Wagenmaker et al.). These works, however, either consider RL-based pretraining (while we focus on BC) or do not consider the question of online finetuning.

## 3 PRELIMINARIES

**Mathematical notation.** Let $\lesssim$ denote inequality up to absolute constants, $\triangle_{\mathcal{X}}$ the simplex over $\mathcal{X}$, and $\mathrm{unif}(\mathcal{X})$ the uniform distribution over $\mathcal{X}$. $\mathbb{I}\{\cdot\}$ denotes the indicator function, $\mathbb{E}^\pi[\cdot]$ the expectation under policy $\pi$ and, unless otherwise noted, $\mathbb{E}[\cdot]$ the expectation over the demonstrator dataset.

**Markov decision processes.** We consider decision-making in the context of episodic, fixed-horizon Markov decision processes (MDPs). An MDP $\mathcal{M}$ is denoted by a tuple $(\mathcal{S}, \mathcal{A}, \{P_h\}_{h=1}^H, P_0, r, H)$, where $\mathcal{S}$ is the set of states, $\mathcal{A}$ the set of actions, $P_h : \mathcal{S} \times \mathcal{A} \to \triangle_{\mathcal{S}}$ the next-state distribution at step $h$, $P_0 \in \triangle_{\mathcal{S}}$ the initial state distribution, $r_h : \mathcal{S} \times \mathcal{A} \to \triangle_{[0,1]}$ the reward distribution, and $H$ the horizon. Interaction with $\mathcal{M}$ proceeds in episodes of length $H$. At step 1, we sample a state $s_1 \sim P_0$, take an action $a_1 \in \mathcal{A}$, receive reward $r_1(s_1, a_1)$, and transition to state $s_2 \sim P_1(\cdot \mid s_1, a_1)$. This continues for $H$ steps until the MDP resets. We let $\mathcal{J}(\pi) := \mathbb{E}^\pi[\sum_{h=1}^H r_h(s_h, a_h)]$ denote the expected reward for policy $\pi$ over one episode. In general, our goal is to maximize $\mathcal{J}(\pi)$.

**Behavioral cloning.** We assume we are given some dataset $\mathfrak{D} = \{(s_1^t, a_1^t, \ldots, s_H^t, a_H^t)\}_{t=1}^T$ collected by running a *demonstrator* policy $\pi^\beta$ on $\mathcal{M}$, so that $(s_1^t, a_1^t, \ldots, s_H^t, a_H^t)$ denotes a full trajectory rollout of $\pi^\beta$ on $\mathcal{M}$, with $a_h^t \sim \pi_h^\beta(\cdot \mid s_h^t)$. We assume that $\pi^\beta$ is Markovian but otherwise make no further assumptions on it (so in particular, $\pi^\beta$ may be stochastic and suboptimal). Our demonstrator dataset does not include reward labels—preventing standard offline RL approaches from applying—but we assume that we have access to reward labels during online interactions.

*Behavioral cloning* (BC) attempts to fit a policy $\widehat{\pi}^\beta$ to match the action distribution of $\pi^\beta$ using $\mathfrak{D}$. Typically this is achieved via supervised learning, where $\widehat{\pi}^\beta$ is trained to predict $a$ given $s$ for $(s, a) \in \mathfrak{D}$. In the tabular setting, which we consider in Section 4, the natural choice for $\widehat{\pi}^\beta$ simply fits the empirical distribution of actions in $\mathfrak{D}$:

$$\widehat{\pi}_h^\beta(a \mid s) := \tfrac{T_h(s,a)}{T_h(s)} \cdot \mathbb{I}\{T_h(s) > 0\} + \mathrm{unif}(\mathcal{A}) \cdot \mathbb{I}\{T_h(s) = 0\} \tag{1}$$

where $T_h(s, a) = \sum_{t=1}^T \mathbb{I}\{(s_h^t, a_h^t) = (s, a)\}$ and $T_h(s) = \sum_{t=1}^T \mathbb{I}\{s_h^t = s\}$. The following result bounds the suboptimality of this estimator, and shows that it is optimal estimator, up to log factors.

**Proposition 1** (Rajaraman et al. (2020)). *If $\mathfrak{D}$ contains $T$ demonstrator trajectories, we have* $\mathcal{J}(\pi^\beta) - \mathbb{E}[\mathcal{J}(\widehat{\pi}^\beta)] \lesssim \frac{H^2 S \log T}{T}$. *Furthermore, for any estimator $\widehat{\pi}$, there exists some MDP $\mathcal{M}$ and demonstrator $\pi^\beta$ such that* $\mathcal{J}(\pi^\beta) - \mathbb{E}[\mathcal{J}(\widehat{\pi})] \gtrsim \min\{H, \frac{H^2 S}{T}\}$.

In other words, without additional reward information, we cannot in general hope to obtain a policy from $\mathfrak{D}$ that does better than (1), if our goal is to maximize the performance of the pretrained policy.

## 4 DEMONSTRATOR ACTION COVERAGE VIA POSTERIOR SAMPLING

In this section we seek to understand how pretraining affects the ability to further improve the downstream policy with RL finetuning, and how we might pretrain to enable downstream improvement. For simplicity, here we assume that our MDP $\mathcal{M}$ is tabular, and let $S$ and $A$ denote the cardinalities of the state and action spaces, respectively; we will show how our proposed approach can be extended to more general settings in the following section.

### 4.1 DEMONSTRATOR ACTION COVERAGE AS A PREREQUISITE FOR FINETUNING

The performance of RL finetuning depends significantly on the RL algorithm applied. Rather than limiting our results to a particular RL algorithm, we instead focus on what is often a prerequisite for effective application of any such approach—demonstrating that the *support* of the pretrained policy is sufficient to enable improvement. In particular, we consider the following definition for the "effective" support of a policy, relative to the demonstrator policy $\pi^\beta$.

**Definition 4.1** ($\gamma$-sampler). We say that policy $\pi$ is a $\gamma$-sampler of $\pi^\beta$ if, for all $(s, h) \in \mathcal{S} \times [H]$ and $a \in \mathcal{A}$, we have that $\pi_h^\beta(a \mid s) \geq \gamma \cdot \pi_h(a \mid s)$.

The majority of RL finetuning approaches rely on rolling out the pretrained policy—which we denote as $\widehat{\pi}^{\mathrm{pt}}$—online, and using the collected observations to finetune its behavior. If our pretrained policy is a $\gamma$-sampler of $\pi^\beta$, then this ensures that any action sampled by $\pi^\beta$ will also be sampled by $\widehat{\pi}^{\mathrm{pt}}$ in these rollouts (with some probability). While this is not a *sufficient* condition for online

improvement, it is a *necessary* condition, in some cases, for performing as well as the demonstrator $\pi^\beta$ (as Proposition 2 demonstrates), and is therefore a necessary condition for improving over $\pi^\beta$. Furthermore, the *value* of $\gamma$ also has impact on the computational cost of RL finetuning. A $\gamma$-sampler requires a factor of $\frac{1}{\gamma}$ more samples than $\pi^\beta$ to ensure it samples some action in the support of $\pi^\beta$. For approaches such as Best-of-$N$ sampling that rely on sampling many actions from the pretrained policy and then taking the best one, a large value of $\gamma$ therefore ensures that we can efficiently sample actions likely to be sampled by the demonstrator policy $\pi^\beta$, while if $\gamma$ is small, it may require taking a significant number of samples from $\widehat{\pi}^{\mathrm{pt}}$ to ensure we cover the behavior of $\pi^\beta$, greatly increasing the computational cost due to this sampling.

In the following, we aim to understand how we can pretrain policies that are $\gamma$-samplers, and to do this with large values of $\gamma$. Furthermore, we aim to achieve this without incurring significant additional suboptimality as compared to $\widehat{\pi}^\beta$—we would like to ensure that $\widehat{\pi}^{\mathrm{pt}}$ is an effective initialization for finetuning while still itself achieving effective online performance.

### 4.2 Behavioral Cloning Fails to Achieve Action Coverage

We first consider standard BC, i.e. (1). The following result shows that the estimator in (1), despite achieving the optimal suboptimality rate, can fail to achieve sufficient action coverage, and that this fundamentally limits its ability to serve as an effective initialization for finetuning.

**Proposition 2** (Informal). *Fix any $\epsilon \in (0, 1/8)$. Then there exists some MDP $\mathcal{M}$ and demonstrator policy $\pi^\beta$ such that, unless $T \geq \frac{1}{20\epsilon}$, we have that, with probability at least $1/2$:*

$$\mathcal{J}(\pi^\beta) - \epsilon > \max_{\pi \in \widehat{\Pi}} \mathcal{J}(\pi) \quad for \quad \widehat{\Pi} := \{\pi : \pi_h(a \mid s) = 0 \text{ if } \widehat{\pi}_h^\beta(a \mid s) = 0, \forall s, a, h\}.$$

*Furthermore, if we collect samples with $\widehat{\pi}^\beta$ on $\mathcal{M}$ we will not be able to identify an $\epsilon$-optimal policy.*

We state the full version of Proposition 2 as Proposition 5 in the appendix. Proposition 2 shows that, unless we have a sufficiently large demonstrator dataset ($T \geq \frac{1}{20\epsilon}$), half of the time (i.e. half of the random draws of the demonstrator dataset) the policy returned by standard BC will not contain a near-optimal policy in its support and, furthermore, that rolling out $\widehat{\pi}^\beta$ on $\mathcal{M}$ will therefore not allow us to learn a near-optimal policy on $\mathcal{M}$. In other words, some fraction of the time standard BC produces a policy which will simply *never* play actions required to solve the task at the level of the demonstrator policy, and any online improvement approach that relies on rolling out the BC pretrained policy to collect observations will therefore fail to identify an $\epsilon$-optimal policy—online improvement is not possible with this pretrained policy. This implies that pretraining a policy that matches the demonstrator's empirical action distribution as represented in $\mathfrak{D}$—the typical goal of behavioral cloning—is insufficient for downstream RL finetuning.

A straightforward solution to this is to simply add exploration noise to our pretrained policy—rather than playing $\widehat{\pi}^\beta$ at every step, with some probability play a random action. While this will clearly address the shortcoming of generative BC outlined above—*every* action will now be in the support—as the following result shows, there is a fundamental tradeoff between the suboptimality of this policy and the number of samples from the policy required to ensure we cover our demonstrator's behavior.

**Proposition 3.** *Fix $T > 0$, $H \geq 2$, $S \geq \lceil \log_2 4T \rceil + 2$, $\xi \geq 0$, define $\epsilon := \frac{H^2 S \log T}{T} + \xi$, and assume $\epsilon \leq \frac{1}{2}$. Define the policy $\widehat{\pi}^{\mathrm{u},\alpha}$ as $\widehat{\pi}_h^{\mathrm{u},\alpha}(\cdot \mid s) := (1-\alpha) \cdot \widehat{\pi}_h^\beta(\cdot \mid s) + \alpha \cdot \mathrm{unif}(\mathcal{A})$. Then there exists some MDP $\mathcal{M}$ with $S$ states, 2 actions, and horizon $H$ where, in order to ensure that:*

*1. $\mathcal{J}(\pi^\beta) - \mathbb{E}[\mathcal{J}(\widehat{\pi}^{\mathrm{u},\alpha})] \leq \epsilon$,*

*2. $\widehat{\pi}^{\mathrm{u},\alpha}$ is a $\gamma$-sampler of $\pi^\beta$ with probability at least $1 - \delta$, for $\delta \in (0, 1/4e)$,*

*we must have $\alpha \leq 32\epsilon$ and $\gamma \leq \frac{64}{A} \cdot \epsilon$. Furthermore, with probability at least $1/4e$, we have*

$$\mathcal{J}(\pi^\beta) - \frac{1}{T} \cdot \epsilon > \max_{\pi \in \widehat{\Pi}} \mathcal{J}(\pi) \quad for \quad \widehat{\Pi} := \{\pi : \pi_h(a \mid s) = 0 \text{ if } \widehat{\pi}_h^\beta(a \mid s) = 0, \forall s, a, h\}.$$

In order to achieve the $\frac{H^2 S \log T}{T}$ suboptimality rate achieved by standard BC, Proposition 3 then shows that we must have $\gamma \lesssim \frac{1}{A} \cdot \frac{H^2 S \log T}{T}$ or, in other words, to ensure we sample a particular action from $\widehat{\pi}^{\mathrm{u},\alpha}$ that is sampled by $\pi^\beta$, it will require sampling a factor of $\frac{AT}{H^2 S \log T}$ *more samples*

from $\widehat{\pi}^{\mathrm{u},\alpha}$ than it would require from $\pi^\beta$. While this does enable approaches like Best-of-$N$ to improve the policy, in settings where $T$ is large, this requires a significant number of samples from the pretrained policy, greatly increasing the computational burden of such an approach. Furthermore, Proposition 3 shows that this limitation is critical—if we seek to shortcut this exploration and set $\alpha \leftarrow 0$, we will fail to match the performance of $\pi^\beta$ on this instance completely.

### 4.3 Demonstrator's Posterior Policy Achieves Action Coverage

Can we do better than this? Here we show that mixing the BC policy with the *posterior* on the demonstrator's policy achieves a near optimal balance between suboptimality and action coverage.

**Definition 4.2** (Posterior Demonstrator Policy). Given prior distribution $P_{\mathrm{prior}}^\beta \in \triangle_\Pi$ over demonstrator policies, let $P_{\mathrm{post}}^\beta(\cdot \mid \mathfrak{D})$ denote the posterior distribution given demonstration dataset $\mathfrak{D}$. We then define the *posterior demonstrator policy* $\widehat{\pi}^{\mathrm{post}}$ as $\widehat{\pi}_h^{\mathrm{post}}(a \mid s) := \mathbb{E}_{\pi \sim P_{\mathrm{post}}^\beta(\cdot \mid \mathfrak{D})}[\pi_h(a \mid s)]$.

$\widehat{\pi}^{\mathrm{post}}$ is the expected policy of the demonstrator under prior $P_{\mathrm{prior}}^\beta$ given observations $\mathfrak{D}$. In practice, we require a slightly regularized version of $\widehat{\pi}^{\mathrm{post}}$, $\widehat{\pi}^{\mathrm{post},\lambda}$, which is identical to $\widehat{\pi}^{\mathrm{post}}$ if $HT \lesssim e^A$, and otherwise adds a small amount of regularization (see Section B.3). We have the following.

**Theorem 1.** *Let $P_{\mathrm{prior}}^\beta$ be the uniform distribution over Markovian policies, and set $\widehat{\pi}^{\mathrm{pt}}$ to*

$$\widehat{\pi}_h^{\mathrm{pt}}(a \mid s) = (1 - \alpha) \cdot \widehat{\pi}_h^\beta(a \mid s) + \alpha \cdot \widehat{\pi}_h^{\mathrm{post},\lambda}(a \mid s) \qquad (2)$$

*for $\alpha = \frac{1}{\max\{A, H, \log(HT)\}}$. Then*

$$\mathcal{J}(\pi^\beta) - \mathbb{E}[\mathcal{J}(\widehat{\pi}^{\mathrm{pt}})] \lesssim \frac{H^2 S \log T}{T},$$

*and with probability at least $1 - \delta$, for all $(s, a, h)$,*

$$\widehat{\pi}_h^{\mathrm{pt}}(a \mid s) \gtrsim \frac{1}{A + H + \log(HT)} \cdot \min\left\{ \frac{\pi_h^\beta(a \mid s)}{\log(SH/\delta)}, \frac{1}{A + \log(HT)} \right\}.$$

**Theorem 2.** *Fix any $A > 1$ and $T > 1$. Then there exists a family of MDPs $\{\mathcal{M}^i\}_{i \in [A]}$ such that each $\mathcal{M}^i$ has $A$ actions and $S = H = 1$, and if any estimator $\widehat{\pi}$ satisfies $\mathcal{J}^{\mathcal{M}^i}(\pi^{\beta,i}) - \mathbb{E}^{\mathcal{M}^i}[\mathcal{J}(\widehat{\pi})] \le c \cdot \frac{H^2 S \log T}{T}$ for all $i \in [A]$ and some constant $c > 0$, then for $\widehat{\pi}$ to be a $\gamma$-sampler of $\pi^{\beta,i}$ on each $\mathcal{M}^i$ with probability at least $\delta \in (0, 1/4)$, we must have $\gamma \le c \cdot \frac{\log T}{A}$.*

Theorem 1 shows that our choice of $\widehat{\pi}^{\mathrm{pt}}$ achieves the same suboptimality guarantee as $\widehat{\pi}^\beta$—it performs no worse that $\widehat{\pi}^\beta$—and requires only a factor of $\approx A + H$ more samples to ensure we sample a particular action from $\pi^\beta$ than $\pi^\beta$ itself does for actions $a$ such that $\pi_h^\beta(a \mid s) \lesssim 1/A$ (and otherwise requires at most a factor of $A(A + H)$ more). Furthermore, Theorem 2 shows that, to achieve this optimal suboptimality guarantee, any estimator *must* take a factor of $A$ more samples than $\pi^\beta$. In other words, if we want a policy that preserves the optimality of $\widehat{\pi}^\beta$ while playing a diverse enough distribution to enable further online improvement, mixing the posterior demonstrator policy with the BC policy achieves the near-optimal tradeoff, and plays all actions taken by $\pi^\beta$ with minimal computational overhead and without incurring additional suboptimality over the BC policy.

## 5 Posterior Behavioral Cloning

We next show this approach can be instantiated in continuous control settings with expressive generative policy classes. To motivate our instantiation, consider the setting where:

$$\pi_h^\beta(\cdot \mid s) = \mathcal{N}(\mu_h(s), \sigma_h^2(s) \cdot I),$$

for (unknown) $\mu_h(s) \in \mathbb{R}^d$ and (known) $\sigma_h(s) \in \mathbb{R}$. Assume we have observations $\mathfrak{D} = \{a_1, \ldots, a_k\} \sim \pi_h^\beta(\cdot \mid s)$ and a $\mathcal{N}(0, I)$ prior on $\mu_h(s)$. The following result, an extension of Osband et al. (2018), shows we can approximate posterior samples by fitting to "noisy" actions.

**Proposition 4.** *We have $P_{\mathrm{post}}^\beta(\cdot \mid \mathfrak{D}) = \mathcal{N}(\frac{1}{\sigma_h^2(s)+k} \cdot \sum_{t=1}^k a_t, \frac{\sigma_h^2(s)}{\sigma_h^2(s)+k} \cdot I)$ and, if we set*

$$\widehat{\mu}_h(s) = \arg\min_\mu \sum_{i=1}^k \|\mu - \widetilde{a}_i\|_2^2 + \sigma_h^2(s) \cdot \|\mu - \widetilde{\mu}_h(s)\|_2^2,$$

*for $\widetilde{a}_t = a_t + w_t$, $w_t \sim \mathcal{N}(0, \sigma_h^2(s) \cdot I)$, and $\widetilde{\mu} \sim \mathcal{N}(0, I)$, then $\widehat{\mu}_h(s) \sim P_{\mathrm{post}}^\beta(\cdot \mid \mathfrak{D})$.*

Proposition 4 shows that we can compute samples from the posterior on $\mu_h(s)$ by simply fitting a "noised" version of our demonstrations. While in practice our data likely does not satisfy this Gaussianity assumption, the above argument nonetheless suggests that a simple approach to capture the behavior of $\widehat{\pi}_h^{\text{post}}(\cdot \mid s)$ is to generate a "noisy" version of $\mathfrak{D}$ by perturbing the actions in $\mathfrak{D}$ with random noise, then fitting some predictor $f$ on this noisy version of $\mathfrak{D}$. By repeating this $K$ times, we can generate $K$ approximate posterior samples $\{f_\ell\}_{\ell \in [K]}$.

Our theory suggests, however, that we should sample not simply from the posterior, but from $\widehat{\pi}^{\text{post}}$, the expected policy under the posterior. In the Gaussian setting of Proposition 4, to sample from $\widehat{\pi}_h^{\text{post}}(\cdot \mid s)$ it suffices to perturb a sample from the posterior, $\widehat{\mu}_h(s)$, by 0-mean noise with the demonstrator's covariance: $\widehat{\mu}_h(s) + w \sim \widehat{\pi}_h^{\text{post}}(\cdot \mid s)$ if $w \sim \mathcal{N}(0, \sigma_h^2(s) \cdot I)$. If we do not know the demonstrator's covariance, we can approximate it by sampling, for $(s, a) \in \mathfrak{D}$: $\widetilde{a} = a + w$ where $w \sim \mathcal{N}(0, \frac{\sigma_h^2(s)}{\sigma_h^2(s)+k} \cdot I)$. Note that the covariance of $a$'s distribution is precisely the demonstrator's covariance, since $a \sim \pi_h^\beta(\cdot \mid s)$. Therefore, $\widetilde{a}$ will be distributed with the demonstrator's mean and covariance, plus 0-mean noise sampled with the posterior's covariance. While the *mean* of this distribution differs from $\widehat{\pi}_h^{\text{post}}(\cdot \mid s)$, its covariance matches the covariance of $\widehat{\pi}_h^{\text{post}}(\cdot \mid s)$. As we show in Lemma 8, the difference in mean between $\widehat{\pi}_h^{\text{post}}(\cdot \mid s)$ and $\pi_h^\beta(\cdot \mid s)$ is distributed approximately as the posterior's covariance, suggesting that the difference in mean between $\widetilde{a}$ and $\widehat{\pi}_h^{\text{post}}(\cdot \mid s)$ is therefore effectively washed out by the posterior's randomness—$\widetilde{a}$ is sampled approximately as $\widehat{\pi}_h^{\text{post}}(\cdot \mid s)$. To produce an approximate sample from $\widehat{\pi}^{\text{post}}(\cdot \mid s)$ in the general case, then, we sample:

$$\widetilde{a} = a + \alpha \cdot w, \quad w \sim \mathcal{N}(0, \text{cov}(s)), \tag{3}$$

for any $(s, a) \in \mathfrak{D}$, and where $\text{cov}(s) := \sum_{\ell=1}^{K}(f_\ell(s) - \bar{f}(s))(f_\ell(s) - \bar{f}(s))^\top$ for $\bar{f}(s) \leftarrow \frac{1}{K}\sum_{\ell=1}^{K} f_\ell(s)$, and $\alpha$ is some weighting we can tune as desired.

### 5.1 POSTERIOR BEHAVIORAL CLONING

Applying Proposition 4 and Equation (3), we can generate approximate samples from $\widehat{\pi}^{\text{post}}(\cdot \mid s)$ for any $s$ in our demonstration dataset. Theorem 1 suggests that, to obtain a pretrained policy $\widehat{\pi}^{\text{pt}}$ that is an effective initialization for RL finetuning, it suffices to fit $\widehat{\pi}^{\text{pt}}$ to a mixture distribution of the BC policy and $\widehat{\pi}^{\text{post}}$. Approximating this mixture by modulating $\alpha$ in (3), we arrive at the following.

---

**Algorithm 1** Posterior Behavioral Cloning (POSTBC)

1: **input:** demonstration dataset $\mathfrak{D}$, generative model class $\widehat{\pi}^\theta$, posterior weight $\alpha$
2: Generate approximate posterior samples $\{f_\ell\}_{\ell \in [K]}$ and compute $\text{cov}(\cdot)$ from $\{f_\ell\}_{\ell \in [K]}$ as above
3: **for** $i = 1, 2, 3, \ldots$ **do**
4:     Draw batch $\mathfrak{D}_i \sim \text{unif}(\mathfrak{D})$
5:     For all $(s, a) \in \mathfrak{D}_i$, compute $\widetilde{a}$ as in (3) using $\text{cov}(\cdot)$ and $\alpha$, and set $\widetilde{\mathfrak{D}}_i \leftarrow \{(s, \widetilde{a}) : s \in \mathfrak{D}\}$
6:     Take gradient step on $\widehat{\pi}^\theta$ on loss of $\widetilde{\mathfrak{D}}_i$

---

With $\widehat{\pi}^\theta$ an expressive generative model, Algorithm 1 will produce a policy which, instead of fitting the empirical distribution of the demonstrator, fits the full expected posterior of the demonstrator's behavior. This approximates the posterior mixture in Equation (2), and, Theorem 1 suggests, leads to a more effective initialization for RL finetuning, instantiating the behavior illustrated in Figure 1. While Proposition 4 motivates a principled method for generating approximate posterior samples, the precise method used to generate such samples is not a critical part of our approach, and any other method to generate posterior samples can also be combined with Algorithm 1. In particular, we find that in many cases computing $f_\ell$ by fitting on a dataset generated by *bootstrapped sampling*— generating a dataset by sampling with replacement from $\mathfrak{D}$ (Fushiki et al., 2005; Osband & Van Roy, 2015; Osband et al., 2016a)—often leads to more effective performance.

## 6 EXPERIMENTS

Finally, we seek to demonstrate that in practice posterior behavioral cloning (a) enables more efficient RL finetuning of pretrained policies, and (b) leads to a pretrained policy that performs effectively itself, on par with the BC pretrained policy. We focus on continuous control domains, in particular robotic control. We test on both the Robomimic (Mandlekar et al., 2021) and Libero

(Liu et al., 2023) simulators. `Robomimic` is comprised of several robotic manipulation tasks, providing a set of human demonstrations on each task, and enables training and finetuning of single-task BC policies. We consider the `Lift`, `Can`, and `Square` tasks on `Robomimic`. `Libero` similarly contains a variety of robotic manipulation tasks with provided human demonstrations, but enables multi-task training, allowing for pretraining on large corpora of data and then finetuning on particular tasks of interest. In particular, we rely on a subset of the `Libero 90` suite of tasks, training and evaluating on the first 21 tasks, corresponding to three different kitchen manipulation scenes. See Figure 2 for a visualization of our settings. Further details on all experiments can be found in Section D and additional ablations can be found in Section D.3.

We instantiate $\widehat{\pi}^{\mathrm{pt}}$ with a diffusion model, which has become the de-facto standard for parameterizing BC policies in continuous control settings (Chi et al., 2023; Ankile et al., 2024a; Dasari et al., 2024; Team et al., 2024; Black et al., 2024; Bjorck et al., 2025). For the `Robomimic` experiments, we use an MLP-based architecture, trained on a single-task demonstration dataset, and rely on state-based observations. For `Libero`, we utilize a diffusion transformer architecture due to Dasari et al. (2024) and rely on image-based observations and language task conditioning. In `Libero`, we pre-train a single $\widehat{\pi}^{\mathrm{pt}}$ policy on the demonstration data from all 21 tasks (Black et al., 2024; Kim et al., 2024; Khazatsky et al., 2024), and then run RL finetuning on each individual task. To leave room for RL improvement (i.e. to ensure performance is not saturated by the pretrained policy) we limit the number of demos per task in the pretraining dataset. In all cases, we use a binary success reward.

In principle, POSTBC can be combined with any RL finetuning algorithm, and we seek to demonstrate that it improves performance on a representative set of approaches. In particular, we consider DSRL (Wagenmaker et al., 2025), which refines a pretrained diffusion policy's distribution by running RL over its latent-noise space, DPPO (Ren et al., 2024), an on-policy policy-gradient-style algorithm for finetuning diffusion policies, and Best-of-$N$ sampling. For DSRL and DPPO we utilize the publicly available implementations without modification. Best-of-$N$ can be instantiated in a variety of ways (see e.g. Chen et al. (2022); Hansen-Estruch et al. (2023); He et al. (2024); Nakamoto et al. (2024); Dong et al. (2025b)). Here we instantiate it by rolling out the pretrained policy on the task of interest $T$ times (where $T$ is specified in our results) to collect trajectories labeled with success and failure, and train a $Q$-function via IQL (Kostrikov et al., 2021) on these trajectories. At test time, we again roll out the pretrained policy but at each state sample $N$ actions from the policy, and play the action that has the largest value under the IQL-trained $Q$-function.

As baselines, we consider running standard BC pretraining on $\mathfrak{D}$, as well as what we refer to as $\sigma$-BC, where instead of perturbing the actions in $\mathfrak{D}$ by the posterior variance as in (3), we instead perturb them by uniform, state-independent noise with variance $\sigma^2$. This is then equivalent to POSTBC, except we set $\mathrm{cov}(s) = \sigma^2 \cdot I$ for some fixed $\sigma > 0$ in (3) (note that this is a continuous analog to the approach considered in Proposition 3). This itself is a novel approach and our theory predicts it too may lead to improved performance over pretraining with standard BC. On `Robomimic`, we also compare against VALUEDICE (Kostrikov et al., 2019) (which we abbreviate as DICE), a imitation learning approach that attempts to learn a policy with state distribution matching the state distribution of the demonstrations, and only requires access to offline demonstration data. For all experiments, error bars denote 1 standard error. All results are averaged over from 3-5 seeds and policies are evaluated with 200 rollouts for `Robomimic` and 100 for `Libero`.

### 6.1 POSTERIOR BC ENABLES EFFICIENT RL FINETUNING

Our results from running DSRL on `Libero` are given in Figure 3 and on `Robomimic` in Figure 4. For `Libero`, we run DSRL on three tasks from scene 2: "open the top drawer of the cabinet", "put the black bowl at the front on the plate", and "put the middle black bowl on the plate". We see that POSTBC pretraining leads to significant gains for `Libero`, enabling efficient RL finetuning in settings where both standard BC pretraining and $\sigma$-BC pretraining fail. On `Robomimic`, POSTBC significantly outperforms both baselines on `Square`, and achieves modest gains over BC on `Lift` and `Can` (requiring roughly $2\times$ fewer samples to achieve 75% performance than BC). Our results for DPPO are given in Figure 4 where we see that POSTBC pretraining again leads to substantial gains on `Square`. This illustrates that POSTBC can improve performance even of on-policy RL-finetuning algorithms that modify the weights of the pretrained policy. We note as well that, even in the cases when POSTBC does not yield substantial gains, it performs no worse than BC.

Our Best-of-$N$ results are given in Table 1. We see that across settings, POSTBC-pretraining leads to consistent improvements over both BC- and $\sigma$-BC-pretrained policies, and also consistently

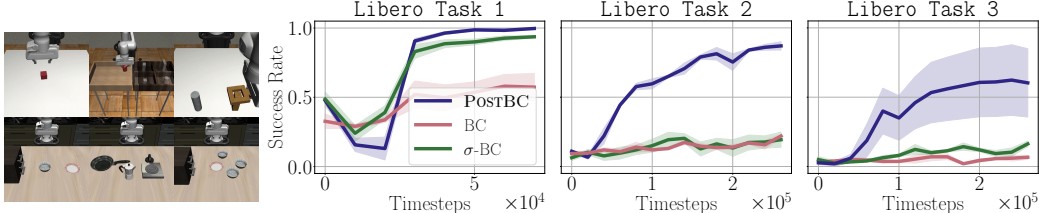

Figure 2: Robomimic and Libero settings

Figure 3: Comparison of DSRL finetuning performance combined with different BC pretraining approaches on Libero.

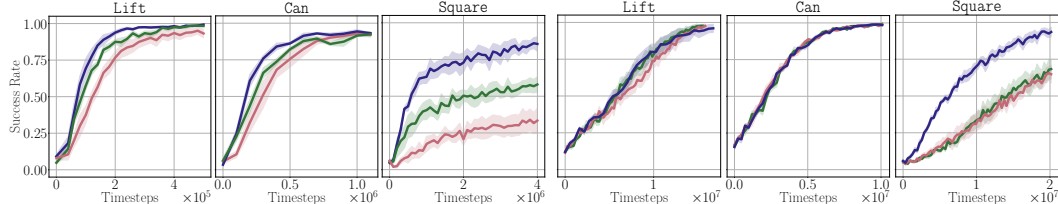

Figure 4: Comparison of DSRL finetuning performance combined with different BC pretraining approaches on Robomimic.

Figure 5: Comparison of DPPO finetuning performance combined with different BC pretraining approaches on Robomimic.

| Task | Pretrained Performance | | Best-of-$N$ (1000 Rollouts) | | | | Best-of-$N$ (2000 Rollouts) | | | |
| | BC | POSTBC | BC | $\sigma$-BC | DICE | POSTBC | BC | $\sigma$-BC | DICE | POSTBC |
| --- | --- | --- | --- | --- | --- | --- | --- | --- | --- | --- |
| Robomimic Lift | **70.1** ±1.7 | 68.1 ±0.7 | 55.6 ±2.4 | 52.3 ±3.7 | 42.3 ±8.6 | 63.3 ±2.1 | 63.8 ±3.6 | 73.5 ±1.1 | 57.8 ±9.0 | **75.7** ±2.0 |
| Robomimic Can | 43.4 ±0.6 | 41.6 ±0.4 | 69.8 ±2.9 | 72.8 ±3.0 | 40.2 ±8.4 | 73.3 ±3.2 | 76.6 ±2.4 | 80.7 ±1.4 | 49.5 ±8.5 | **79.5** ±1.9 |
| Robomimic Square | **18.8** ±0.3 | 17.7 ±0.3 | 37.9 ±2.3 | 45.7 ±1.4 | 11.6 ±1.9 | 48.3 ±1.2 | 48.4 ±1.0 | 50.0 ±3.2 | 18.5 ±1.9 | **52.4** ±2.4 |
| Libero Scene 1 | 22.1 ±8.3 | 24.4 ±6.1 | 38.0 ±7.2 | 63.9 ±3.8 | - | 60.8 ±4.5 | 47.0 ±6.4 | 66.8 ±4.3 | - | **76.3** ±3.0 |
| Libero Scene 2 | 11.5 ±3.4 | 13.1 ±3.9 | 21.7 ±3.6 | 26.7 ±5.0 | - | 44.4 ±5.7 | 23.9 ±4.2 | 29.7 ±4.5 | - | **48.4** ±4.4 |
| Libero Scene 3 | 40.1 ±10.4 | 42.0 ±10.2 | 49.2 ±7.0 | 51.8 ±7.1 | - | 65.5 ±6.8 | 51.6 ±10.2 | 59.4 ±7.2 | - | **66.4** ±7.3 |
| Libero All | 22.2 ±4.3 | 23.0 ±3.9 | 33.5 ±3.5 | 43.7 ±3.6 | - | 54.6 ±3.5 | 38.0 ±3.7 | 48.7 ±3.4 | - | **61.6** ±3.0 |

Table 1: Comparison of success rates of pretrained policies and Best-of-$N$ sampling on Robomimic and Libero, for different pretraining approaches. Bolded text denotes best approach. Please see Table 3 for pretrained performance of $\sigma$-BC and DICE.

outperforms VALUEDICE. In particular, on Libero, POSTBC improves by approximately 20% over BC, and 10% over $\sigma$-BC. Table 1 also provides the performance of the pretrained policies, where we see that, in general, the POSTBC-pretrained policy performs on par with the BC-pretrained policy, demonstrating that POSTBC-pretraining produces a policy which performs as well as the BC pretrained policy. Together these results show that in realistic continuous control settings, pretraining with POSTBC can lead to significant improvements over standard BC pretraining in terms of RL finetuning performance, without sacrificing the performance of the pretrained policy itself.

**Understanding how POSTBC improves RL finetuning performance.** Finally, we seek to provide insight into how POSTBC improves RL finetuning performance. In particular, we aim to disambiguate the role of the additional *exploration* a POSTBC policy may provide over a BC policy, versus the role that having access to a larger action distribution at test time might play. While these factors are intimately coupled for DSRL and DPPO, for Best-of-$N$ sampling we can decouple them by selecting the rollout policy (the "exploration" policy) that collects data to learn the filtering function, and the policy whose actions we filter with the learned function at test-time (the "steering" policy).

We consider mixing the role of the BC and POSTBC policy on Robomimic Lift in this way, and provide our results in Table 2. We find that the choice of rollout policy has little impact on performance, but the steering policy can impact perfor-

| BC rollouts + BC steering | BC rollouts + POSTBC steering | POSTBC rollouts + BC steering | POSTBC rollouts + POSTBC steering |
| --- | --- | --- | --- |
| 63.8 ±3.6 | **78.6** ±1.4 | 65.0 ±4.4 | **75.7** ±2.0 |

Table 2: Best-of-$N$ sampling on Robomimic Lift, varying the rollout policy and the steering policy.

mance significantly. This suggests that the utility of POSTBC is primarily in its ability to provide a wider range of actions that can be sampled from the pretrained policy, enabling RL finetuning approaches to easily select the maximizing action.

## REPRODUCIBILITY STATEMENT

Full proofs for all theoretical results are given in the appendix, allowing our results to be checked. For the experimental results, we have stated hyperparameters used in the appendix, and plan to also release our code publicly.

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

# A  ADDITIONAL RELATED WORK

**Reinforcement Learning-Based Pretraining.**  In the RL literature, two lines of work bear some resemblance to ours as well. The *offline-to-online RL* setting aims to train policies with RL on offline datasets that can then be improved with further online interaction (Lee et al., 2022; Ghosh et al., 2022; Kumar et al., 2022; Zhang et al., 2023; Uchendu et al., 2023; Zheng et al., 2023; Ball et al., 2023; Nakamoto et al., 2023), and the *meta-RL* setting aims to meta-learn a policy on some set of tasks which can then be quickly adapted to a new task (Wang et al., 2016; Duan et al., 2016; Finn et al., 2017a; 2018). While similar to our work in that these works also aim to learn behaviors that can be efficiently improved online, the settings differ significantly in that the offline- or meta-pretraining typically requires reward labels (rather than unlabeled demonstrations) and are performed with RL (rather than BC)—in contrast, we study how BC-like pretraining (as noted, the workhorse of most modern applications) can enable efficient online adaptation.

# B  PROOFS

Some algebra shows that in the tabular setting, under the uniform prior, we have

$$\widehat{\pi}_h^{\text{post}}(a \mid s) := \begin{cases} \frac{T_h(s,a)+1}{T_h(s)+A} & T_h(s) > 0 \\ \text{unif}(\mathcal{A}) & o.w. \end{cases}$$

## B.1  BC POLICY FAILS TO COVER ACTIONS

**Proposition 5** (Full version of Proposition 2). *Fix any $\epsilon \in (0, 1/8]$. Then there exist some MDPs $\mathcal{M}^1, \mathcal{M}^2$ and demonstrator policy $\pi^\beta$ such that, if $\mathcal{M} \in \{\mathcal{M}^1, \mathcal{M}^2\}$, unless $T \geq \frac{1}{20\epsilon}$, we have that, with probability at least $1/2$:*

$$\mathcal{J}(\pi^\beta) - \epsilon > \max_{\pi \in \widehat{\Pi}} \mathcal{J}(\pi) \quad for \quad \widehat{\Pi} := \{\pi : \pi_h(a \mid s) = 0 \text{ if } \widehat{\pi}_h^\beta(a \mid s) = 0, \forall s, a, h\}.$$

*Furthermore,*

$$\min_{\widehat{\pi}} \max_{i \in \{1,2\}} \mathbb{E}^{\mathcal{M}^i, \widehat{\pi}^\beta}[\max_\pi \mathcal{J}^{\mathcal{M}^i}(\pi) - \mathcal{J}^{\mathcal{M}^i}(\widehat{\pi})] \geq \frac{1}{2}.$$

*Proof.* Let $\mathcal{M}^1$ and $\mathcal{M}^2$ denote multi-armed bandits with 3 arms and reward functions $r^1$ and $r^2$:

$$r^1(a_1) = 0, r^1(a_2) = 1, r^1(a_3) = 0$$
$$r^2(a_1) = 0, r^2(a_2) = 0, r^2(a_3) = 1.$$

Let $\pi^\beta(a_1) = 1 - 4\epsilon$, $\pi^\beta(a_2) = 2\epsilon$, $\pi^\beta(a_3) = 2\epsilon$.

By construction of $\widehat{\pi}^\beta$, if $T(a_2) = 0$ then we will have $\widehat{\pi}^\beta(a_2) = 0$, and if $T(a_3) = 0$ we will have $\widehat{\pi}^\beta(a_3) = 0$. By the definition of both $\mathcal{M}^1$ and $\mathcal{M}^2$, we have

$$\mathbb{P}^{\mathcal{M}^i}[T(a_2) = 0, T(a_3) = 0] = (1 - 4\epsilon)^T.$$

As we have assumed that $T \leq \frac{1}{20\epsilon}$ and $\epsilon \in (0, 1/8]$, some calculation shows that we can lower bound this as $1/2$. Note that for both $\mathcal{M}^1$ and $\mathcal{M}^2$, we have $\mathcal{J}(\pi^\beta) = 2\epsilon$, while for policies $\widehat{\pi}^\beta$ that only play $a_1$, we have $\mathcal{J}(\widehat{\pi}^\beta) = 0$. This proves the first part of the result.

For the second part, note that the optimal policy on $\mathcal{M}^1$ plays only $a_2$ and has expected reward of 1, while the optimal policy on $\mathcal{M}^2$ plays only $a_2$ and has expected reward of 1. Let $\widehat{\pi}$ denote an estimate of the optimal policy and $\mathbb{E}^{\mathcal{M}^i, \widehat{\pi}^\beta}[\cdot]$ the expectation induced by playing the policy $\widehat{\pi}^\beta$ from the first part on instance $\mathcal{M}^i$. Then:

$$\min_{\widehat{\pi}} \max_{i \in \{1,2\}} \mathbb{E}^{\mathcal{M}^i, \widehat{\pi}^\beta}[\max_\pi \mathcal{J}^{\mathcal{M}^i}(\pi) - \mathcal{J}^{\mathcal{M}^i}(\widehat{\pi})] = \min_{\widehat{\pi}} \max_{i \in \{1,2\}} \mathbb{E}^{\mathcal{M}^i, \widehat{\pi}^\beta}[1 - \widehat{\pi}(a_{1+i})].$$

Note that $1 - \widehat{\pi}(a_2) = \widehat{\pi}(a_1) + \widehat{\pi}(a_3) \geq \widehat{\pi}(a_3)$. Thus we can lower bound the above as

$$\geq \min_{\widehat{\pi}} \max\{\mathbb{E}^{\mathcal{M}^1, \widehat{\pi}^\beta}[\widehat{\pi}(a_3)], \mathbb{E}^{\mathcal{M}^2, \widehat{\pi}^\beta}[1 - \widehat{\pi}(a_3)]\}$$

$$\geq \min_{\widehat{\pi}} \frac{1}{2} \left( \mathbb{E}^{\mathcal{M}^1, \widehat{\pi}^\beta}[\widehat{\pi}(a_3)] + \mathbb{E}^{\mathcal{M}^2, \widehat{\pi}^\beta}[1 - \widehat{\pi}(a_3)] \right)$$

$$\geq \frac{1}{2} - \frac{1}{2} \min_{\widehat{\pi}} \left| \mathbb{E}^{\mathcal{M}^1, \widehat{\pi}^\beta}[\widehat{\pi}(a_3)] - \mathbb{E}^{\mathcal{M}^2, \widehat{\pi}^\beta}[\widehat{\pi}(a_3)] \right|.$$

We can bound

$$\left| \mathbb{E}^{\mathcal{M}^1, \widehat{\pi}^\beta}[\widehat{\pi}(a_3)] - \mathbb{E}^{\mathcal{M}^2, \widehat{\pi}^\beta}[\widehat{\pi}(a_3)] \right| \leq \mathrm{TV}(\mathbb{P}^{\mathcal{M}^1, \widehat{\pi}^\beta}, \mathbb{P}^{\mathcal{M}^2, \widehat{\pi}^\beta}).$$

Since $\mathcal{M}^1$ and $\mathcal{M}^2$ only differ on $a_2$ and $a_3$, and since $\widehat{\pi}^\beta(a_2) = \widehat{\pi}^\beta(a_3) = 0$, we have $\mathrm{TV}(\mathbb{P}^{\mathcal{M}^1, \widehat{\pi}^\beta}, \mathbb{P}^{\mathcal{M}^2, \widehat{\pi}^\beta}) = 0$. Thus, we conclude that

$$\min_{\widehat{\pi}} \max_{i \in \{1,2\}} \mathbb{E}^{\mathcal{M}^i, \widehat{\pi}^\beta}[\max_\pi \mathcal{J}^{\mathcal{M}^i}(\pi) - \mathcal{J}^{\mathcal{M}^i}(\widehat{\pi})] \geq \frac{1}{2}.$$

This proves the second part of the result.

$\square$

## B.2 UNIFORM NOISE FAILS

*Proof of Proposition 3.* **Construction.** Let $\mathcal{M}$ be the MDP with state space $\{\widetilde{s}_1, \ldots, \widetilde{s}_k, s_1, s_2\}$, actions $\{a_1, a_2\}$, horizon $H \geq 2$ with initial state distribution:

$$P_0(s_1) = 1/2, \quad P_0(\widetilde{s}_1) = 2^{-2} + 2^{-k}, \quad P_0(\widetilde{s}_i) = 2^{-i-1}, i \geq 2,$$

transition function, for all $h \in [H]$:

$$P_h(\widetilde{s}_i \mid \widetilde{s}_i, a) = 1, \forall a \in \mathcal{A}, \quad P_h(s_1 \mid s_1, a_1) = 1,$$
$$P_h(s_2 \mid s_1, a_2) = 1, \quad P_h(s_2 \mid s_2, a) = 1, \forall a \in \mathcal{A},$$

and reward that is 0 everywhere except

$$r_1(\widetilde{s}_i, a_1) = r_H(s_1, a_1) = 1, \quad r_1(\widetilde{s}_i, a_2) = 1 - 2\Delta,$$

for some $\Delta > 0$ to be specified. We consider $\pi^\beta$ defined as

$$\pi_h^\beta(a_1 \mid \widetilde{s}_i) = \pi_h^\beta(a_2 \mid \widetilde{s}_i) = \frac{1}{2}, \quad \pi_h^\beta(a_1 \mid s_1) = 1.$$

Let $\epsilon := \frac{H^2 S \log T}{T} + \xi$, and set $\Delta \leftarrow 2\epsilon$.

**Upper bound on $\alpha$.** Note that $\mathcal{J}(\pi^\beta) = 1 - \frac{1}{2}\Delta$, and that the value of the optimal policy $\pi^\star$ is $\mathcal{J}(\pi^\star) = \max_\pi \mathcal{J}(\pi) = 1$. Let $\widetilde{\pi}^{\mathrm{u},\alpha}$ denote the policy that, on all $\widetilde{s}_i$ plays $\pi^\star$, and on other states plays $\pi^\star$ with probability $1 - \alpha$, and otherwise plays $\mathrm{unif}(\mathcal{A})$. Note then that, regardless of the value of $\widehat{\pi}^\beta$, we have that $\mathcal{J}(\widetilde{\pi}^{\mathrm{u},\alpha}) \geq \mathcal{J}(\widehat{\pi}^{\mathrm{u},\alpha})$. Thus,

$$\mathcal{J}(\pi^\beta) - \mathbb{E}[\mathcal{J}(\widehat{\pi}^{\mathrm{u},\alpha})] \geq \mathcal{J}(\pi^\beta) - \mathcal{J}(\widetilde{\pi}^{\mathrm{u},\alpha})$$

If we are in $s_1$ at $h = 2$, the only way we can receive any reward on the episode is if we take action $a_1$ for the last $H - 1$ steps, and we then receive a reward of 1. Under $\widetilde{\pi}^{\mathrm{u},\alpha}$, we take $a_1$ at each step with probability $1 - \alpha + \alpha/A$, so our probability of getting a reward of 1 is $(1 - \alpha + \alpha/A)^{H-1}$. Note that in contrast $\pi^\beta$ will always play $a_1$ and receive a reward of 1 in this situation. If we are in $\widetilde{s}_i$ at $h = 2$ for any $i$, then $\pi^\beta$ will incur a loss of $\Delta$ more than $\widetilde{\pi}^{\mathrm{u},\alpha}$. Thus, we can lower bound

$$\mathcal{J}(\pi^\beta) - \mathcal{J}(\widetilde{\pi}^{\mathrm{u},\alpha}) \geq -\frac{1}{2}\Delta + \frac{1}{2} \cdot (1 - (1 - \alpha + \alpha/A)^{H-1})$$

By assumption we have that $\frac{1}{2}\Delta = \epsilon$. Thus, if we want $\mathcal{J}(\pi^\beta) - \mathbb{E}[\mathcal{J}(\widehat{\pi}^{\mathrm{u},\alpha})] \leq \epsilon$, we need

$$\frac{1}{2} \cdot (1 - (1 - \alpha + \alpha/A)^{H-1}) \leq 2\epsilon.$$

Rearranging this, we have

$$1 - 4\epsilon \le (1 - \alpha + \alpha/A)^{H-1} \iff \frac{1}{H-1} \log{(1 - 4\epsilon)} \le \log(1 - \alpha + \alpha/A).$$

From the Taylor decomposition of $\log(1 - x)$, we see that $\log(1 - \alpha + \alpha/A) \le -(1 - 1/A)\alpha$. Furthermore, we can lower bound

$$\log(1 - 4\epsilon) \ge -8\epsilon$$

as long as $\epsilon \le 1/2$. Altogether, then, we have

$$\frac{-8\epsilon}{H-1} \le -(1 - 1/A)\alpha \implies \alpha \le \frac{8\epsilon}{(H-1)(1 - 1/A)} \implies \alpha \le 32\epsilon$$

where the last inequality follows since $H \ge 2, A = 2$.

**Upper bound on $\gamma$.** Let $i_T := \arg\max_i\{2^{-i-1} \mid 2^{-i-1} \le 1/T\}$, so that $1/2T \le P_0(\widetilde{s}_{i_T}) \le 1/T$, and note that such an $\widetilde{s}_{i_T}$ exists by construction. Let $\mathcal{E}$ be the event $\mathcal{E} := \{T_1(\widetilde{s}_{i_T}) = T_1(\widetilde{s}_{i_T}, a_2) = 1\}$. We have

$$\begin{aligned}
\mathbb{P}[\mathcal{E}] &= \mathbb{P}[T_1(\widetilde{s}_{i_T}, a_2) = 1 \mid T_1(\widetilde{s}_{i_T}) = 1]\mathbb{P}[T_1(\widetilde{s}_{i_T}) = 1] \\
&= \frac{1}{2} \cdot TP_0(\widetilde{s}_{i_T})(1 - P_0(\widetilde{s}_{i_T}))^{T-1} \\
&= \frac{1}{2} \cdot T \cdot \frac{1}{2T} \cdot (1 - \frac{1}{T})^{T-1} \\
&\ge \frac{1}{4e}.
\end{aligned}$$

Note that on the event $\mathcal{E}$, we have $\widehat{\pi}_1^\beta(a_1 \mid \widetilde{s}_{i_T}) = 0$, but $\pi_1^\beta(a_1 \mid \widetilde{s}_{i_T}) = 1/2$. Thus,

$$\widehat{\pi}_1^{\mathrm{u},\alpha}(a_1 \mid \widetilde{s}_{i_T}) = \alpha/A \le 32\epsilon/A = 64\epsilon/A \cdot \pi_1^\beta(a_1 \mid \widetilde{s}_{i_T})$$

where we have used the bound on $\alpha$ shown above. Thus, on $\mathcal{E}$, we will only have that $\widehat{\pi}^{\mathrm{u},\alpha}$ is a $\gamma$-sampler for $\gamma \le 64\epsilon/A$. Since $\mathcal{E}$ occurs with probability at least $1/4e$, it follows that if we want to guarantee $\widehat{\pi}^{\mathrm{u},\alpha}$ is a $\gamma$-sampler with probability at least $1 - \delta$ for $\delta < 1/4e$, we must have $\gamma \le 64\epsilon/A$.

Note as well that, since $\widehat{\pi}_1^\beta(a_2 \mid \widetilde{s}_{i_T}) = 1$, any policy in the support of $\widehat{\pi}^\beta$ will be suboptimal by a factor of at least $P_0(\widetilde{s}_{i_T}) \cdot 2\Delta \ge \Delta/T$. $\qquad\square$

### B.3 ANALYSIS OF POSTERIOR POLICY

Throughout this section we denote

$$\widetilde{\pi}_h(a \mid s) := \begin{cases} (1 - \alpha) \cdot \frac{T_h(s,a)}{T_h(s)} + \alpha \cdot \frac{T_h(s,a) + \lambda/A}{T_h(s) + \lambda} & T_h(s) > 0 \\ \mathrm{unif}(\mathcal{A}) & T_h(s) = 0 \end{cases}$$

for some $\alpha \in [0, 1]$.

We also denote $w_h^\pi(s, a) := \mathbb{P}^\pi[s_h = s, a_h = a]$. $Q_h^\pi(s, a) := \mathbb{E}^\pi[\sum_{h' \ge h} r_{h'}(s_{h'}, a_{h'}) \mid s_h = s, a_h = a]$ denotes the standard $Q$-function. $\mathcal{J}(\pi; r)$ denotes the expected return of policy $\pi$ for reward $r$.

**Lemma 1.** *As long as $\delta \le 0.9$ and $\lambda \ge A$, we have*

$$\mathbb{P}\left[\widetilde{\pi}_h(a \mid s) \ge \alpha \cdot \min\left\{\frac{\pi_h^\beta(a \mid s)}{64 \log SH/\delta}, \frac{1}{2\lambda}\right\}, \forall a \in \mathcal{A}, s \in \mathcal{S}, h \in [H]\right] \ge 1 - \delta.$$

*Proof.* Consider some $(s, h)$. By Bernstein's inequality, if $T_h(s) > 0$, we have that with probability at least $1 - \delta$,

$$\frac{T_h(s,a)}{T_h(s)} \ge \pi_h^\beta(a \mid s) - \sqrt{\frac{2\pi_h^\beta(a \mid s) \log 1/\delta}{T_h(s)}} - \frac{2 \log 1/\delta}{3T_h(s)}. \qquad (4)$$

From some algebra, we see that as long as $T_h(s) \geq \frac{32 \log 1/\delta}{\pi_h^\beta(a|s)}$, we have that $\frac{T_h(s,a)}{T_h(s)} \geq \frac{1}{2}\pi_h^\beta(a \mid s)$. By the definition of $\widetilde{\pi}$, under the good event of (4) we can then lower bound

$$
\widetilde{\pi}_h(a \mid s) \geq \begin{cases} \frac{\alpha}{1+\lambda/T_h(s)} \cdot \frac{1}{2}\pi_h^\beta(a \mid s) & T_h(s) \geq \frac{32 \log 1/\delta}{\pi_h^\beta(a|s)} \\ \frac{\alpha\lambda/A}{T_h(s)+A} & \text{o.w.} \end{cases}
$$

$$
\geq \begin{cases} \frac{\alpha \cdot 32 \log 1/\delta}{32 \log 1/\delta + \lambda \cdot \pi_h^\beta(a|s)} \cdot \frac{1}{2}\pi_h^\beta(a \mid s) & N_h(s) \geq \frac{32 \log 1/\delta}{\pi_h^\beta(a|s)} \\ \frac{\alpha\lambda/A \cdot \pi_h^\beta(a|s)}{32 \log 1/\delta + \lambda \cdot \pi_h^\beta(a|s)} & \text{o.w.} \end{cases}
$$

$$
\overset{(a)}{\geq} \frac{\alpha \cdot \pi_h^\beta(a \mid s)}{32 \log 1/\delta + \lambda \cdot \pi_h^\beta(a \mid s)}
$$

$$
\geq \alpha \cdot \min\left\{ \frac{\pi_h^\beta(a \mid s)}{64 \log 1/\delta}, \frac{1}{2\lambda} \right\}
$$

where $(a)$ follows as long as $\delta \leq 0.9$ and $\lambda \geq A$. In the case when $T_h(s) = 0$ we have $\widetilde{\pi}_h(a \mid s) = 1/A \geq 1/\lambda$, so this lower bound still holds. Taking a union bound over arms proves the result. $\square$

**Lemma 2.** *As long as $\lambda \geq 4\log(HT)$, we have*

$$
\mathbb{E}[\mathcal{J}(\widehat{\pi}^\beta) - \mathcal{J}(\widetilde{\pi})] \lesssim (1 + \alpha H) \cdot \frac{H^2 S \log T}{T} + \alpha \cdot \frac{H^2 S\lambda}{T}.
$$

*Proof.* By the Performance-Difference Lemma we have:

$$
\mathcal{J}(\widehat{\pi}^\beta) - \mathcal{J}(\widetilde{\pi}) = \sum_{h=1}^H \sum_{s \in \mathcal{S}} w_h^{\widehat{\pi}^\beta}(s) \cdot \left( \mathbb{E}_{a \sim \widehat{\pi}_h^\beta(s)}[Q_h^{\widetilde{\pi}}(s,a)] - \mathbb{E}_{a \sim \widetilde{\pi}_h(s)}[Q_h^{\widetilde{\pi}}(s,a)] \right)
$$

$$
\leq \sum_{h=1}^H \sum_{s \in \mathcal{S}} w_h^{\widehat{\pi}^\beta}(s) \cdot \left| \mathbb{E}_{a \sim \widehat{\pi}_h^\beta(s)}[Q_h^{\widetilde{\pi}}(s,a)] - \mathbb{E}_{a \sim \widetilde{\pi}_h(s)}[Q_h^{\widetilde{\pi}}(s,a)] \right|. \qquad (5)
$$

For $(s,h)$ with $N_h(s) > 0$, we have

$$
\left| \mathbb{E}_{a \sim \widehat{\pi}_h^\beta(s)}[Q_h^{\widetilde{\pi}}(s,a)] - \mathbb{E}_{a \sim \widetilde{\pi}_h(s)}[Q_h^{\widetilde{\pi}}(s,a)] \right| \leq \sum_{a \in \mathcal{A}} H \cdot |\widehat{\pi}_h^\beta(a \mid s) - \widetilde{\pi}_h(a \mid s)|,
$$

where we have used that $Q_h^{\widehat{\pi}^{\mathrm{post}}}(s,a) \in [0, H]$. Then, using the definition of $\widehat{\pi}^\beta$ and $\widetilde{\pi}$ we can bound this as

$$
\leq \sum_{a \in \mathcal{A}} \alpha H \cdot \left| \frac{T_h(s,a)}{T_h(s)} - \frac{T_h(s,a) + \lambda/A}{T_h(s) + \lambda} \right|
$$

$$
= \sum_{a \in \mathcal{A}} \frac{\alpha\lambda H}{A} \cdot \left| \frac{AT_h(s,a) - T_h(s)}{T_h(s)(T_h(s) + \lambda)} \right|
$$

$$
\leq \sum_{a \in \mathcal{A}} \frac{\alpha\lambda H}{A} \cdot \frac{AT_h(s,a) + T_h(s)}{T_h(s)(T_h(s) + \lambda)}
$$

$$
= \frac{2\alpha\lambda H}{T_h(s) + \lambda}.
$$

Since $\mathbb{E}_{a \sim \widehat{\pi}_h^\beta(s)}[Q_h^{\widetilde{\pi}}(s,a)] - \mathbb{E}_{a \sim \widetilde{\pi}_h(s)}[Q_h^{\widetilde{\pi}}(s,a)] = 0$ by construction when $T_h(s) = 0$, we then have

$$
(5) \leq \sum_{h=1}^H \sum_{s \in \mathcal{S}} w_h^{\widehat{\pi}^\beta}(s) \cdot \frac{2\alpha\lambda H}{T_h(s) + \lambda}.
$$

Let $\mathcal{E}$ denote the good event from Lemma 3 with $\delta = \frac{S}{T}$. Then as long as $\lambda \geq 4\log(HT)$ we can bound the above as

$$\leq \sum_{h=1}^{H} \sum_{s \in \mathcal{S}} w_h^{\widehat{\pi}^\beta}(s) \cdot \frac{2\alpha\lambda H}{T_h(s) + \lambda} \mathbb{I}\{\mathcal{E}\} + 2H^2 \cdot \mathbb{I}\{\mathcal{E}^c\}$$

$$\leq \sum_{h=1}^{H} \sum_{s \in \mathcal{S}} w_h^{\widehat{\pi}^\beta}(s) \cdot \frac{4\alpha\lambda H}{w_h^{\pi^\beta}(s) \cdot T + \lambda} + 2H^2 \cdot \mathbb{I}\{\mathcal{E}^c\}.$$

Let $\widetilde{r}$ denote the reward function:

$$\widetilde{r}_h(s, a) := \frac{\lambda}{w_h^{\pi^\beta}(s) \cdot T + \lambda}$$

and note that $\widetilde{r} \in [0, 1]$, and

$$\sum_{h=1}^{H} \sum_{s \in \mathcal{S}} w_h^{\widehat{\pi}^\beta}(s) \cdot \frac{4\alpha\lambda H}{w_h^{\pi^\beta}(s) \cdot T + \lambda} = 4\alpha H \cdot \mathcal{J}(\widehat{\pi}^\beta; \widetilde{r}).$$

By Theorem 4.4 of Rajaraman et al. (2020), we have[1]

$$\mathbb{E}[\mathcal{J}(\widehat{\pi}^\beta; \widetilde{r})] \lesssim \mathcal{J}(\pi^\beta; \widetilde{r}) + \frac{H^2 S \log T}{T}$$

$$= \sum_{h=1}^{H} \sum_{s \in \mathcal{S}} w_h^{\pi^\beta}(s) \cdot \frac{\lambda}{w_h^{\pi^\beta}(s) \cdot T + \lambda} + \frac{H^2 S \log T}{T}$$

$$\leq \frac{HS\lambda}{T} + \frac{H^2 S \log T}{T}.$$

Noting that $\mathbb{E}[2H^2 \cdot \mathbb{I}\{\mathcal{E}^c\}] \leq 2H^2\delta \leq \frac{2H^2 S}{T}$ completes the proof. $\qquad\square$

**Lemma 3.** *With probability at least $1 - \delta$, for all $(s, h)$, we have*

$$T_h(s) + \lambda \geq \frac{1}{2}w_h^{\pi^\beta}(s) \cdot T + \frac{1}{2}\lambda$$

*as long as $\lambda \geq 4\log\frac{SH}{\delta}$.*

*Proof.* Consider some $(s, h)$ and note that $\mathbb{E}[T_h(s)/T] = w_h^{\pi^\beta}(s)$. By Bernstein's inequality, we have with probability $1 - \delta/SH$:

$$T_h(s) \geq w_h^{\pi^\beta}(s) \cdot T - \sqrt{2w_h^{\pi^\beta}(s) \cdot T \cdot \log\frac{SH}{\delta}} - \frac{2}{3}\log\frac{SH}{\delta}.$$

We would then like to show that

$$w_h^{\pi^\beta}(s) \cdot T - \sqrt{2w_h^{\pi^\beta}(s) \cdot T \cdot \log\frac{SH}{\delta}} - \frac{2}{3}\log\frac{SH}{\delta} + \lambda \geq \frac{1}{2}(w_h^{\pi^\beta}(s) \cdot T + \lambda)$$

$$\iff \frac{1}{2}w_h^{\pi^\beta}(s) \cdot T + \frac{1}{2}\lambda \geq \sqrt{2w_h^{\pi^\beta}(s) \cdot T \cdot \log\frac{SH}{\delta}} + \frac{2}{3}\log\frac{SH}{\delta}$$

As we have assumed $\lambda \geq 4\log\frac{SH}{\delta}$, it suffices to show

$$\frac{1}{2}w_h^{\pi^\beta}(s) \cdot T + \log\frac{SH}{\delta} \geq \sqrt{2w_h^{\pi^\beta}(s) \cdot T \cdot \log\frac{SH}{\delta}}.$$

However, this is true by the AM-GM inequality. A union bound proves the result. $\qquad\square$

---

[1]Note that Theorem 4.4 of Rajaraman et al. (2020) shows an inequality in the opposite direction of what we show here: they bound $\mathcal{J}(\pi^\beta; \widetilde{r}) - \mathbb{E}[\mathcal{J}(\widehat{\pi}^\beta; \widetilde{r})]$ instead of $\mathbb{E}[\mathcal{J}(\widehat{\pi}^\beta; \widetilde{r})] - \mathcal{J}(\pi^\beta; \widetilde{r})$. However, we see that the only place in their proof where their argument relied on this ordering is in Lemma A.8. We show in Lemma 4 that a reverse version of their Lemma A.8 holds, allowing us to instead bound $\mathbb{E}[\mathcal{J}(\widehat{\pi}^\beta; \widetilde{r})] - \mathcal{J}(\pi^\beta; \widetilde{r})$.

**Lemma 4** (Reversed version of Lemma A.8 of Rajaraman et al. (2020)). *Adopting the notation from Rajaraman et al. (2020), we have*

$$\mathbb{E}[\mathrm{Pr}_{\pi^{\mathrm{first}}}[\mathcal{E}]] \leq \frac{SH \log N}{N}$$

*for $\mathcal{E}^c$ the event that within a trajectory, the policy only visits states for which $T_h(s) > 0$.*

*Proof.* Let $\mathcal{E}_{s,h}$ denote the event that the state $s$ is visited at step $h$ and $T_h(s) = 0$, and $\mathcal{E}_h := \cup_{s \in \mathcal{S}} \mathcal{E}_{s,h}$. Then, by simple set inclusions, we have:

$$\mathcal{E} = \bigcup_{h \in [H]} \bigcup_{s \in \mathcal{S}} \mathcal{E}_{s,h} = \bigcup_{h \in [H]} \bigcup_{s \in \mathcal{S}} \left( \mathcal{E}_{s,h} \cap \bigcap_{h' < h} \mathcal{E}_{h'}^c \right).$$

By a union bound it follows that

$$\mathbb{E}[\mathrm{Pr}_{\pi^{\mathrm{first}}}[\mathcal{E}]] \leq \sum_{h \in [H]} \sum_{s \in \mathcal{S}} \mathbb{E}[\mathrm{Pr}_{\pi^{\mathrm{first}}}[\mathcal{E}_{s,h} \cap \bigcap_{h' < h} \mathcal{E}_{h'}^c]].$$

Now note that

$$\mathrm{Pr}_{\pi^{\mathrm{first}}}[\mathcal{E}_{s,h} \cap \bigcap_{h' < h} \mathcal{E}_{h'}^c] = \mathrm{Pr}_{\pi^{\mathrm{first}}}[\mathcal{E}_{s,h} \mid \bigcap_{h' < h} \mathcal{E}_{h'}^c] \mathrm{Pr}_{\pi^{\mathrm{first}}}[\bigcap_{h' < h} \mathcal{E}_{h'}^c]$$

$$= \mathrm{Pr}_{\pi^{\mathrm{first}}}[\mathcal{E}_{s,h} \mid \bigcap_{h' < h} \mathcal{E}_{h'}^c] \mathrm{Pr}_{\pi^{\mathrm{first}}}[\mathcal{E}_{h-1}^c \mid \bigcap_{h' < h-1} \mathcal{E}_{h'}^c] \mathrm{Pr}_{\pi^{\mathrm{first}}}[\bigcap_{h' < h-1} \mathcal{E}_{h'}^c]$$

$$\vdots$$

$$= \mathrm{Pr}_{\pi^{\mathrm{first}}}[\mathcal{E}_{s,h} \mid \bigcap_{h' < h} \mathcal{E}_{h'}^c] \cdot \prod_{h' < h} \mathrm{Pr}_{\pi^{\mathrm{first}}}[\mathcal{E}_{h'}^c \mid \bigcap_{h'' < h'} \mathcal{E}_{h''}^c].$$

If the event $\bigcap_{h' < h} \mathcal{E}_{h'}^c$ holds, then up to step $h$ no states are encountered for which $T_{h'}(s) = 0$. Thus, on such states, $\pi^{\mathrm{first}}$ and $\pi^{\mathrm{orc-first}}$ will behave identically. It follows that $\mathbb{E}[\mathrm{Pr}_{\pi^{\mathrm{first}}}[\mathcal{E}_{s,h} \mid \bigcap_{h' < h} \mathcal{E}_{h'}^c]] = \mathbb{E}[\mathrm{Pr}_{\pi^{\mathrm{orc-first}}}[\mathcal{E}_{s,h} \mid \bigcap_{h' < h} \mathcal{E}_{h'}^c]]$. By a similar argument, we have $\mathrm{Pr}_{\pi^{\mathrm{orc-first}}}[\mathcal{E}_{h'}^c \mid \bigcap_{h'' < h'} \mathcal{E}_{h''}^c] = \mathrm{Pr}_{\pi^{\mathrm{first}}}[\mathcal{E}_{h'}^c \mid \bigcap_{h'' < h'} \mathcal{E}_{h''}^c]$ for each $h' < h$. Thus,

$$\mathrm{Pr}_{\pi^{\mathrm{first}}}[\mathcal{E}_{s,h} \cap \bigcap_{h' < h} \mathcal{E}_{h'}^c] = \mathrm{Pr}_{\pi^{\mathrm{orc-first}}}[\mathcal{E}_{s,h} \cap \bigcap_{h' < h} \mathcal{E}_{h'}^c].$$

It follows that

$$\mathbb{E}[\mathrm{Pr}_{\pi^{\mathrm{first}}}[\mathcal{E}]] \leq \sum_{h \in [H]} \sum_{s \in \mathcal{S}} \mathbb{E}[\mathrm{Pr}_{\pi^{\mathrm{orc-first}}}[\mathcal{E}_{s,h} \cap \bigcap_{h' < h} \mathcal{E}_{h'}^c]] \leq \sum_{h \in [H]} \sum_{s \in \mathcal{S}} \mathbb{E}[\mathrm{Pr}_{\pi^{\mathrm{orc-first}}}[\mathcal{E}_{s,h}]].$$

From here the proof follows identically to the proof of Lemma A.8 of Rajaraman et al. (2020). $\square$

*Proof of Theorem 1.* Set $\lambda = \max\{A, 4 \log(HT)\}$ and $\alpha = \frac{1}{\max\{A, H, \log(HT)\}}$. We have

$$\mathcal{J}(\pi^\beta) - \mathbb{E}[\mathcal{J}(\widehat{\pi}^\beta)] + \mathbb{E}[\mathcal{J}(\widehat{\pi}^\beta)] - \mathbb{E}[\mathcal{J}(\widetilde{\pi})] \lesssim \frac{H^2 S \log T}{T} + (1 + \alpha H) \cdot \frac{H^2 S \log T}{T} + \alpha \cdot \frac{H^2 S \lambda}{T}$$

where we bound $\mathcal{J}(\pi^\beta) - \mathbb{E}[\mathcal{J}(\widehat{\pi}^\beta)]$ by Theorem 4.4 of Rajaraman et al. (2020), and $\mathbb{E}[\mathcal{J}(\widehat{\pi}^\beta)] - \mathbb{E}[\mathcal{J}(\widetilde{\pi})]$ by Lemma 2 since $\lambda \geq 4 \log(HT)$. By our choice of $\alpha = \frac{1}{\max\{A, H, \log(HT)\}}$, we can bound all of this as

$$\lesssim \frac{H^2 S \log T}{T}.$$

This proves the suboptimality guarantee. To show that $\widetilde{\pi}$ is a $\gamma$-sampler, we applying Lemma 1 using our values of $\lambda$ and $\alpha$ $\square$

### B.4 OPTIMALITY OF POSTERIOR SAMPLING

Let $\mathcal{M}$ denote a multi-armed bandit with $A$ actions where $r(a_1) = 1$ and $r(a_i) = 0$ for $i > 1$. Let $\pi^{\beta,i}$ denote the policy defined as

$$\pi^{\beta,i}(a) = \begin{cases} 1 - \alpha & a = 1 \\ \alpha & a = i \\ 0 & \text{o.w.} \end{cases}$$

for $i > 1$ and $\alpha$ some value we will set, and $\pi^{\beta,1}(1) = 1$. We let $\mathcal{M}^i = (\mathcal{M}, \pi^{\beta,i})$ the instance-demonstrator pair, $\mathbb{E}^i[\cdot]$ the expectation on this instance, $\mathbb{P}^i$ the distribution on this instance, and $\mathbb{P}^{i,T} = \otimes_{t=1}^T \mathbb{P}^i$.

**Lemma 5.** *Consider the instance constructed above. Then we have that, for $j \neq i$:*

$$\mathbb{P}^i[\widehat{\pi}(i) \geq \gamma \cdot \alpha] \leq 2 \cdot \mathbb{P}^j[\widehat{\pi}(i) \geq \gamma \cdot \alpha] + T \cdot \alpha.$$

*Proof.* This follows from Lemma A.11 of Foster et al. (2021), which immediately gives that:

$$\mathbb{P}^i[\{\widehat{\pi}(i) \geq \gamma \cdot \alpha] \leq 2 \cdot \mathbb{P}^j[\widehat{\pi}(i) \geq \gamma \cdot \alpha] + D_{\mathrm{H}}^2(\mathbb{P}^{i,T}, \mathbb{P}^{j,T}),$$

where $D_{\mathrm{H}}(\cdot, \cdot)$ denotes the Hellinger distance. Since the squared Hellinger distance is subadditive we have

$$D_{\mathrm{H}}^2(\mathbb{P}^{i,T}, \mathbb{P}^{j,T}) \leq T \cdot D_{\mathrm{H}}^2(\mathbb{P}^i, \mathbb{P}^j).$$

By elementary calculations we see that $D_{\mathrm{H}}^2(\mathbb{P}^i, \mathbb{P}^j) = \alpha$, which proves the result. $\square$

**Theorem 3** (Full version of Theorem 2). *Let $\widehat{\pi}$ be a $\gamma$-sampler of $\pi^\beta$ for each $\mathcal{M}^i, i \in [A]$, and some $\delta \in (0, 1/4]$, and assume that*

$$\mathcal{J}(\pi^{\beta,i}) - \mathbb{E}^i[\mathcal{J}(\widehat{\pi})] \leq \xi, \quad \forall i \geq 1$$

*for some $\xi > 0$. Then if $T \leq \frac{1}{4\alpha}$, it must be the case that*

$$\gamma \leq \frac{\xi}{4A\alpha}.$$

*In particular, setting $\xi = c \cdot \frac{\log T}{T}$ and if $\alpha = \frac{1}{4T}$, we have*

$$\gamma \leq c \cdot \frac{\log T}{A}.$$

*Proof.* Our goal is to find the maximum value of $\gamma$ such that our constraint on the optimality of $\widehat{\pi}$ is met, for each $\mathcal{M}^i$. In particular, this can be upper bounded as

$$\max_{\widehat{\pi}, \gamma} \gamma \quad \text{s.t.} \quad \mathbb{P}^i[\{\widehat{\pi}(a) \geq \gamma \cdot \pi^\beta(a), \forall a \in \mathcal{A}\}] \geq 1 - \delta, \ \mathcal{J}(\pi^{\beta,i}) - \mathbb{E}^i[\mathcal{J}(\widehat{\pi})] \leq \xi, \ \forall i \geq 1. \quad (6)$$

Note that for $\mathcal{M}^i, i \geq 1$, the event $\{\widehat{\pi}(a) \geq \gamma \cdot \pi^{\beta,i}(a), \forall a \in \mathcal{A}\}$ is a subset of the event $\{\widehat{\pi}(i) \geq \gamma \cdot \alpha\}$. This allows us to bound (6) as

$$\max_{\widehat{\pi}, \gamma} \gamma \quad \text{s.t.} \quad \mathbb{P}^i[\widehat{\pi}(i) \geq \gamma \cdot \alpha] \geq 1 - \delta, \ \mathcal{J}(\pi^{\beta,i}) - \mathbb{E}^i[\mathcal{J}(\widehat{\pi})] \leq \xi, \ \forall i \geq 1. \quad (7)$$

By Lemma 5, we have that for each $i > 1$,

$$\mathbb{P}^i[\widehat{\pi}(i) \geq \gamma \cdot \alpha] \leq 2 \cdot \mathbb{P}^1[\widehat{\pi}(i) \geq \gamma \cdot \alpha] + T \cdot \alpha.$$

Furthermore, on $\mathcal{M}^1$ we have $\mathcal{J}(\pi^{\beta,1}) - \mathbb{E}^1[\mathcal{J}(\widehat{\pi})] = \mathbb{E}^1[\sum_{i>1} \widehat{\pi}(i)]$. Given this, we can upper bound (7) as

$$\max_{\widehat{\pi}, \gamma} \gamma \quad \text{s.t.} \quad \mathbb{P}^1[\widehat{\pi}(i) \geq \gamma \cdot \alpha] \geq \frac{1}{2} \cdot (1 - \delta - T \cdot \alpha), \forall i > 1, \ \mathbb{E}^1[\sum_{i>1} \widehat{\pi}(i)] \leq \xi. \quad (8)$$

By Markov's inequality, we have

$$\mathbb{P}^1[\widehat{\pi}(i) \geq \gamma \cdot \alpha] \leq \frac{\mathbb{E}^1[\widehat{\pi}(i)]}{\gamma \cdot \alpha}.$$

Furthermore, since we have assumed $\delta \leq 1/4$ and $T \leq \frac{1}{4\alpha}$, we have $\frac{1}{2} \cdot (1 - \delta - T \cdot \alpha) \geq \frac{1}{4}$. We can therefore bound (8) as

$$\max_{\widehat{\pi}, \gamma} \gamma \quad \text{s.t.} \quad \mathbb{E}^1[\widehat{\pi}(i)] \geq \frac{1}{4} \cdot \gamma\alpha, \forall i > 1, \ \mathbb{E}^1[\sum_{i>1} \widehat{\pi}(i)] \leq \xi. \tag{9}$$

However, we see then that we immediately have that

$$\gamma \leq \frac{\xi}{4A\alpha}.$$

This proves the result. □

## C  APPROXIMATE POSTERIOR

Let $P(\cdot \mid \mu)$ denote the distribution $\mathcal{N}(\mu, \Sigma)$, where we assume $\mu$ is unknown and $\Sigma$ is known. Assume that we have samples $\mathfrak{D} = \{x_1, \ldots, x_T\} \sim P(\cdot \mid \mu^\star)$. Let $Q_{\text{prior}} = \mathcal{N}(0, \Lambda_0)$ denote the prior on $\mu$. Throughout this section we let $=^d$ denote equality in distribution.

**Lemma 6.** *Under $Q_{\text{prior}}$, we have that the posterior $Q_{\text{post}}$ on $\mu$ is:*

$$Q_{\text{post}}(\cdot \mid \mathfrak{D}) = \mathcal{N}\left(\Lambda_{\text{post}}\Sigma^{-1} \cdot \sum_{t=1}^T x_t, \Lambda_{\text{post}}\right),$$

*for $\Lambda_{\text{post}}^{-1} = \Lambda_0^{-1} + T \cdot \Sigma^{-1}$.*

*Proof.* Dropping terms that do not depend on $\mu$, we have

$$Q_{\text{post}}(\mu \mid \mathfrak{D}) = \frac{P(\mathfrak{D} \mid \mu)Q_{\text{prior}}(\mu)}{P(\mathfrak{D})}$$

$$\propto \exp\left(-\frac{1}{2}\sum_{t=1}^T (x_t - \mu)^\top \Sigma^{-1}(x_t - \mu)\right) \cdot \exp\left(-\frac{1}{2}\mu^\top \Lambda_0 \mu\right)$$

$$\propto \exp\left(-\frac{1}{2}T\mu^\top \Sigma^{-1}\mu - \frac{1}{2}\mu^\top Q_{\text{prior}}^{-1}\mu + \mu^\top \Sigma^{-1} \cdot \sum_{t=1}^T x_t\right)$$

$$= \exp\left(-\frac{1}{2}(\mu - \Lambda_{\text{post}}v)^\top \Lambda_{\text{post}}^{-1}(\mu - \Lambda_{\text{post}}v) + \frac{1}{2}v^\top \Lambda_{\text{post}}v\right)$$

for $\Lambda_{\text{post}}^{-1} = \Lambda_0^{-1} + T \cdot \Sigma^{-1}$, and $v = \Sigma^{-1} \cdot \sum_{t=1}^T x_t$. □

**Lemma 7** (General version of Proposition 4). *Let*

$$\widehat{\mu} = \arg\min_\mu \sum_{t=1}^T (\mu - \widetilde{x}_t)^\top \Sigma^{-1}(\mu - \widetilde{x}_t) + (\mu - \widetilde{\mu})^\top \Lambda_0^{-1}(\mu - \widetilde{\mu}),$$

*for $\widetilde{x}_t = x_t + w_t$, $w_t \sim \mathcal{N}(0, \Sigma)$, and $\widetilde{\mu} \sim Q_{\text{prior}}$. Then $\widehat{\mu} =^d Q_{\text{post}}(\cdot \mid \mathfrak{D})$.*

*Proof.* By computing the gradient of the objective, setting it equal to 0, and solving for $\mu$, we see that

$$\widehat{\mu} = (\Lambda_0^{-1} + T\Sigma^{-1})^{-1} \cdot \left(\Sigma^{-1} \cdot \sum_{t=1}^T \widetilde{x}_t + \Lambda_0^{-1}\widetilde{\mu}\right)$$

$$= (\Lambda_0^{-1} + T\Sigma^{-1})^{-1} \cdot \Sigma^{-1} \cdot \sum_{t=1}^T x_t + (\Lambda_0^{-1} + T\Sigma^{-1})^{-1} \cdot \left(\Sigma^{-1} \cdot \sum_{t=1}^T w_t + \Lambda_0^{-1}\widetilde{\mu}\right).$$

Note that the first term in the above is deterministic conditioned on $\mathfrak{D}$, and the second term is mean 0 and has covariance $(\Lambda_0^{-1} + T\Sigma^{-1})^{-1}$. We see then that the mean and covariance of $\widehat{\mu}$ match the mean the covariance of $Q_{\text{post}}(\cdot \mid \mathfrak{D})$ given in Lemma 6, which proves the result. $\square$

**Lemma 8.** *Let $\widetilde{x}$ be distributed as*

$$\widetilde{x} \sim \mathcal{N}(\widehat{\mu}, \Sigma) \quad for \quad \widehat{\mu} \sim Q_{\text{post}}(\cdot \mid \mathfrak{D}) \quad and \quad \mathfrak{D} \sim P(\cdot \mid \mu^\star).$$

*Then*

$$\widetilde{x} =^d x_{T+1} + 2w + z$$

*for $x_{T+1} \sim P(\cdot \mid \mu^\star)$, $w \sim \mathcal{N}(0, \Lambda_{\text{post}})$, and $z$ some random variable satisfying $\mathbb{E}[\|z\|_2^2] \leq \mathcal{O}(1/T^2)$.*

*Proof.* Note that $x_t = \mu^\star + \eta_t$, for $\eta_t \sim \mathcal{N}(0, \Sigma)$. We then have

$$\mu^\star - \Lambda_{\text{post}}\Sigma^{-1} \cdot \sum_{t=1}^{T} x_t = \mu^\star - T\Lambda_{\text{post}}\Sigma^{-1}\mu^\star - \Lambda_{\text{post}}\Sigma^{-1} \cdot \sum_{t=1}^{T} \eta_t. \tag{10}$$

Note that

$$T\Lambda_{\text{post}}\Sigma^{-1}\mu^\star = \Lambda_{\text{post}}(T\Sigma^{-1} + \Lambda_0^{-1})\mu^\star - \Lambda_{\text{post}}\Lambda_0^{-1}\mu^\star = \mu^\star - \Lambda_{\text{post}}\Lambda_0^{-1}\mu^\star.$$

Furthermore, we have that

$$-\Lambda_{\text{post}}\Sigma^{-1} \cdot \sum_{t=1}^{T} \eta_t =^d \mathcal{N}(0, T\Lambda_{\text{post}}\Sigma^{-1}\Lambda_{\text{post}}) =^d \mathcal{N}(0, \Lambda_{\text{post}} - \Lambda_{\text{post}}\Lambda_0^{-1}\Lambda_{\text{post}}).$$

It follows that

$$(10) =^d \mathcal{N}\left(\Lambda_{\text{post}}\Lambda_0^{-1}\mu^\star, \Lambda_{\text{post}} - \Lambda_{\text{post}}\Lambda_0^{-1}\Lambda_{\text{post}}\right).$$

Note that by construction, $\Lambda_{\text{post}}\Lambda_0^{-1}\mu^\star \leq \mathcal{O}(1/T)$. Furthermore, $\|\Lambda_{\text{post}}\Lambda_0^{-1}\Lambda_{\text{post}}\|_2 = \mathcal{O}(1/T^2)$. Thus,

$$(10) =^d \mathcal{N}\left(0, \Lambda_{\text{post}} - \mathcal{O}(1/T^2)\right) + \mathcal{O}^d(1/T)$$

where here we let $\mathcal{O}^d(1/T)$ denote some term $X$ such that $\mathbb{E}[\|X\|_2^2] \leq \mathcal{O}(1/T)$. As a perturbation of $\mathcal{O}(1/T^2)$ to the covariance will result in a perturbation whose norm is bounded in expectation as $\mathcal{O}(1/T)$, we have

$$(10) =^d \mathcal{N}\left(0, \Lambda_{\text{post}}\right) + \mathcal{O}^d(1/T).$$

Let $w \sim \mathcal{N}(0, \Lambda_{\text{post}})$ and $\eta \sim \mathcal{N}(0, \Sigma)$. Then, by Lemmas 6 and 7:

$$\begin{aligned}
\widehat{\mu} + \eta &=^d \Lambda_{\text{post}}\Sigma^{-1} \cdot \sum_{t=1}^{T} x_t + w + \eta \\
&=^d \mu^\star + \mathcal{N}(0, \Lambda_{\text{post}}) + w + \eta + \mathcal{O}^d(1/T) \\
&=^d \mu^\star + 2w + \eta + \mathcal{O}^d(1/T) \\
&=^d x_{T+1} + 2w + \mathcal{O}^d(1/T)
\end{aligned}$$

for $x_{T+1} \sim P(\cdot \mid \mu^\star)$. $\square$

# D  ADDITIONAL EXPERIMENTAL DETAILS

We summarize our approach for generating approximate posterior samples in Algorithm 2. In all experiments, we parameterize $f_\ell$ with Gaussian policy. While using more expressive generative policies to produce the final policy leads to better performance, as we only use $f_\ell$ to estimate the variance at each point, a Gaussian policy suffices. Furthermore, Gaussian policies are often easier to fit than generative policies—often requiring less gradient steps than, for example, diffusion policies—so using a Gaussian policy reduces the computation required as well.

---

**Algorithm 2** Posterior Variance Approximation via Ensembled Prediction

---

1: **input:** demonstration dataset $\mathfrak{D}$, ensemble size $K$, function class $\mathcal{F}$, dataset type ($\in$ {noisy, bootstrapped})
2: **for** $\ell = 1, 2, \ldots, K$ **do**
3:     **if** dataset type == noisy **then**
4:         Set $\mathfrak{D}_\ell \leftarrow \{(s, a + w_{sa}^\ell) : \forall (s, a) \in \mathfrak{D}\}$ where $w_{sa}^\ell \sim \mathcal{N}(0, I)$
5:     **else if** dataset type == bootstrapped **then**
6:         Set $\mathfrak{D}_\ell \leftarrow \{\, |\mathfrak{D}|$ points $(s, a)$ sampled with replacement from $\mathfrak{D}\}$
7:     Fit $f_\ell$ by solving $f_\ell \leftarrow \arg\min_{f \in \mathcal{F}} \sum_{(s,\widetilde{a}) \in \mathfrak{D}_\ell} \|f_\ell(s) - \widetilde{a}\|_2^2$
8: **return** $\{f_\ell\}_{\ell \in [K]}$

---

| Task | Pretrained Performance | | | |
| --- | --- | --- | --- | --- |
| | BC | $\sigma$-BC | DICE | POSTBC |
| Robomimic Lift | **70.1** ±1.7 | 66.7 ±0.8 | 20.0 ±2.4 | 68.1 ±0.7 |
| Robomimic Can | 43.4 ±0.6 | **44.3** ±0.9 | 14.1 ±2.8 | 41.6 ±0.4 |
| Robomimic Square | **18.8** ±0.3 | 18.3 ±0.3 | 6.2 ±0.6 | 17.7 ±0.3 |
| Libero Scene 1 | 22.1 ±8.3 | 23.2 ±6.2 | - | **24.4** ±6.1 |
| Libero Scene 2 | 11.5 ±3.4 | 10.3 ±4.1 | - | **13.1** ±3.9 |
| Libero Scene 3 | 40.1 ±10.4 | 37.4 ±7.6 | - | **42.0** ±10.2 |
| Libero All | 22.2 ±4.3 | 21.1 ±3.7 | - | **23.0** ±3.9 |

Table 3: Comparison of success rates of all pretrained policies on Robomimic and Libero, for different pretraining approaches. Bolded text denotes best approach.

## D.1 ROBOMIMIC EXPERIMENTS

For all Robomimic experiments, we run POSTBC as stated in Algorithm 1 however, instead of computing the full covariance of the posterior, we only compute the diagonal covariance. We instantiate $\widehat{\pi}^\theta$ with a diffusion policy that uses an MLP architecture. For $f_\ell$, we train an MLP to simply predict the noised action directly in $\mathfrak{D}_i$ (i.e. we do not use a diffusion model for $f_\ell$), but use the same architecture and dimensions for $f_\ell$ as the diffusion policies. We used bootstrapped sampling to compute the ensemble for all settings but Best-of-$N$ on Lift. In all cases we pretrain on the Multi-Human Robomimic datasets, and in cases where we use less than the full dataset, we randomly select trajectories from the dataset to train on, using the same trajectories for each approach.

For each RL finetuning method, we sweep over the same hyperparameters for each pretrained policy method (i.e. BC, $\sigma$-BC, POSTBC), and include results for the best one. For $\sigma$-BC, we swept over values of $\sigma$ and included results for the best-performing one. With the exception of DSRL Square, for every Robomimic experiment, we train 5 diffusion policies per pertraining method, and perform a single RL finetuning run on it, so that each stated values is averaged over 5 seeds; For DSRL Square we only average over 3 seeds. For each evaluation, we roll out the policy 200 times. For DPPO we utilize the default hyperparameters as stated in Ren et al. (2024), and utilize DDPM sampling. For VALUEDICE, we use the officially published codebase, and the default hyperparameters provided there. In all cases, we utilize a -1/0 success reward, using Robomimic's built-in success detector to determine the reward. We provide hyperparameters for the individual experiments below.

Table 4: **Common DSRL hyperparameters for all experiments.**

| Hyperparameter | Value |
| --- | --- |
| Learning rate | 0.0003 |
| Batch size | 256 |
| Activation | Tanh |
| Target entropy | 0 |
| Target update rate ($\tau$) | 0.005 |
| Number of actor and critic layers | 3 |
| Number of critics | 2 |
| Number of environments | 4 |

Table 5: DSRL hyperparameters for `Robomimic` experiments.

| Hyperparameter | Lift | Can | Square |
|---|---|---|---|
| Hidden size | 2048 | 2048 | 2048 |
| Gradient steps per update | 20 ($\sigma$-BC), 10 (BC,POSTBC) | 20 | 10 (POSTBC), 20 (BC, $\sigma$-BC) |
| Noise critic update steps | 10 | 10 | 10 |
| Discount factor | 0.99 | 0.99 | 0.999 |
| Action magnitude | 1.5 | 1.5 | 1.5 |
| Initial steps | 24000 | 24000 | 32000 |

Table 6: Hyperparameters for pretrained policies for `Robomimic` DSRL experiments.

| Hyperparameter | Lift | Can | Square |
|---|---|---|---|
| Dataset size (number trajectories) | 5 | 10 | 40 |
| Action chunk size | 4 | 4 | 4 |
| train denoising steps | 100 | 20 | 100 |
| inference denoising steps | 8 | 8 | 8 |
| Hidden size | 512 | 1024 | 1024 |
| Hidden layers | 3 | 3 | 3 |
| Training epochs | 3000 | 3000 | 3000 |
| Ensemble size (POSTBC) | 100 | 10 | 100 |
| Ensemble training epochs (POSTBC) | 10000 | 6000 | 3000 |
| Posterior noise weight $\alpha$ (POSTBC) | 1 | 0.5 | 1 |
| Uniform noise $\sigma$ ($\sigma$-BC) | 0.1 | 0.05 | 0.05 |

Table 7: Best-of-$N$ hyperparameters for `Robomimic` experiments.

| Hyperparameter | Lift | Can | Square |
|---|---|---|---|
| Total gradient steps | 3000000 | 2000000 | 2000000 |
| IQL $\tau$ (1000 rollouts) | 0.7 | 0.7 (BC, $\sigma$-BC, DICE), 0.9 (POSTBC) | 0.7 |
| IQL $\tau$ (2000 rollouts) | 0.7 (BC, $\sigma$-BC, DICE), 0.9 (POSTBC) | 0.7 | 0.7 (BC, $\sigma$-BC, DICE), 0.9 (POSTBC) |
| Discount factor | 0.999 | 0.999 | 0.999 |

Table 8: **Hyperparameters for pretrained policies for `Robomimic` Best-of-$N$ experiments.**

| Hyperparameter | Lift | Can | Square |
|---|---|---|---|
| Dataset size (number trajectories) | 20 | 300 | 300 |
| Action chunk size | 1 | 1 | 1 |
| train denoising steps | 100 | 20 | 100 |
| inference denoising steps | 8 | 8 | 8 |
| Hidden size | 512 | 1024 | 1024 |
| Hidden layers | 3 | 3 | 3 |
| Training epochs | 3000 | 3000 | 3000 |
| Ensemble size (POSTBC) | 10 | 10 | 10 |
| Ensemble noise $\sigma$ (POSTBC) | 0.5 | - | - |
| Ensemble training epochs (POSTBC) | 500 | 500 | 500 |
| Posterior noise weight $\alpha$ (POSTBC) | 2 | 1 | 1 |
| Uniform noise $\sigma$ ($\sigma$-BC) | 0.1 | 0.05 | 0.05 |

Table 9: **Hyperparameters for pretrained policies for `Robomimic` DPPO experiments.**

| Hyperparameter | Lift | Can | Square |
|---|---|---|---|
| Dataset size (number trajectories) | 5 | 10 | 30 |
| Action chunk size | 4 | 4 | 4 |
| train denoising steps | 100 | 100 | 100 |
| Hidden size | 512 | 1024 | 1024 |
| Hidden layers | 3 | 3 | 3 |
| Training epochs | 3000 | 3000 | 3000 |
| Ensemble size (POSTBC) | 100 | 100 | 10 |
| Ensemble training epochs (POSTBC) | 3000 | 6000 | 3000 |
| Posterior noise weight $\alpha$ (POSTBC) | 0.5 | 0.25 | 1 |
| Uniform noise $\sigma$ ($\sigma$-BC) | 0.1 | 0.05 | 0.05 |

## D.2 LIBERO EXPERIMENTS

For Libero, we utilize the transformer architecture from Dasari et al. (2024) for $\widehat{\pi}^\theta$. We run POSTBC as stated in Algorithm 1, but instead of approximating the posterior by adding noise to actions, we instead used a bootstrap estimate, where we sample from $\mathfrak{D}$ with replacement, and fit $f_\ell$ to the boot-strapped samples (we note that this is another common strategy for uncertainty estimation in RL, see e.g. Osband et al. (2016a)). For $f_\ell$, we utilize the same ResNet and tokenizer as $\widehat{\pi}^\theta$, but simply utilize a 3-layer MLP head on top of it—trained to predict the actions directly—rather than a full diffusion transformer. For the Best-of-$N$ experiments, POSTBC utilizes a diagonal posterior covariance estimate, while for the DSRL runs it is trained with the full matrix posterior covariance estimate. We train on Libero-90 data from the first 3 scenes of Libero-90—KITCHEN-SCENE1, KITCHEN-SCENE2, and KITCHEN-SCENE3—and use 25 trajectories from each task in each scene. For task conditioning, we conditioning $\widehat{\pi}^\theta$ on the BERT language embedding (Devlin et al., 2019) of the corresponding text given for that task in the Libero dataset.

For each RL finetuning method, we sweep over the same hyperparameters for each pretrained policy method (i.e. BC, $\sigma$-BC, POSTBC), and include results for the best one. For $\sigma$-BC, we swept over values of $\sigma$ and included results for the best-performing one. The DSRL experiments are averaged over 3 different pretraining runs per method, and one DSRL run per pretrained run. The Best-of-$N$ experiments are averaged over 2 different pretraining runs per method, and 2 Best-of-$N$ runs per pretrained run. For each evaluation, we roll out the policy 100 times. In all cases, we utilize a -1/0 success reward, using Libero's built-in success detector to determine the reward.

We provide hyperparameters for the individual experiments below.

Table 10: **DSRL hyperparameters for all Libero experiments.**

| Hyperparameter | Value |
| --- | --- |
| Learning rate | 0.0003 |
| Batch size | 256 |
| Activation | Tanh |
| Target entropy | 0 |
| Target update rate ($\tau$) | 0.005 |
| Number of actor and critic layers | 3 |
| Layer size | 1024 |
| Number of critics | 2 |
| Number of environments | 1 |
| Gradient steps per update | 20 |
| Discount factor | 0.99 |
| Action magnitude | 1.5 |
| Initial episode rollouts | 20 |

Table 11: **Best-of-$N$ hyperparameters for all Libero experiments.**

| Hyperparameter | Value |
| --- | --- |
| IQL learning rate | 0.0003 |
| IQL batch size | 256 |
| IQL $\beta$ | 3 |
| Activation | Tanh |
| Target update rate | 0.005 |
| $Q$ and $V$ number of layers | 2 |
| $Q$ and $V$ layer size | 256 |
| Number of critics | 2 |
| $N$ (Best-of-$N$ samples) | 32 |
| IQL gradient steps | 50000 |
| IQL $\tau$ | 0.9 |
| Discount factor | 0.99 |

Table 12: **Hyperparameters for DiT diffusion policy in Libero experiments.**

| Hyperparameter | Value |
|---|---|
| Batch size | 150 |
| Learning rate | 0.0003 |
| Training steps | 50000 |
| LR scheduler | cosine |
| Warmup steps | 2000 |
| Action chunk size | 4 |
| Train denoising steps | 100 |
| Inference denoising steps | 8 |
| Image encoder | ResNet-18 |
| Hidden size | 256 |
| Number of Heads | 8 |
| Number of Layers | 4 |
| Feedforward dimension | 512 |
| Token dimension | 256 |
| Ensemble size (POSTBC) | 5 |
| Ensemble training steps (POSTBC) | 25000 |
| Ensemble layer size | 512 |
| Ensemble number of layers | 3 |
| Posterior noise weight (POSTBC) | 2 (DSRL run), 4 (Best-of-$N$ run) |
| Uniform noice $\sigma$ ($\sigma$-BC) | 0.05 |

## D.3 ADDITIONAL ABLATIONS

We provide several additional ablations on POSTBC.

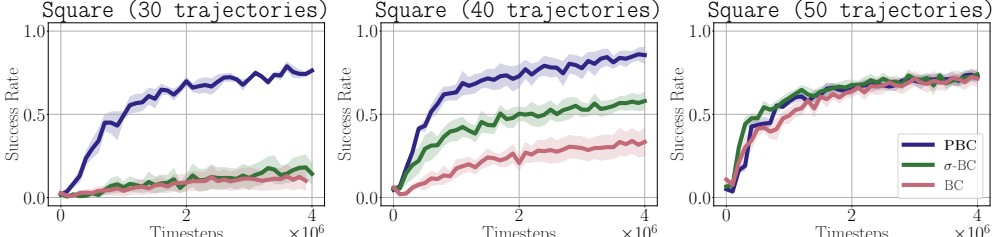

Figure 6: Comparison of DSRL finetuning performance combined with different BC pretraining approaches on `Robomimic Square`, varying the number of trajectories in the dataset the policies are pretrained on. As can be seen, the finetuning performance of policies pretrained with POSTBC is largely unaffected by the size of the pretraining dataset, while BC and $\sigma$-BC are both very sensitive to dataset size. For large enough datasets (50 trajectories), BC and $\sigma$-BC perform as well as POSTBC. This is to be expected—if we train on enough data, our uncertainty will be low, so POSTBC will essentially reduce to BC. These results illustrate that POSTBC gracefully interpolates between settings where BC overfits to small amounts of data, hurting its finetuning performance, and settings where BC is sufficient for effective finetuning.

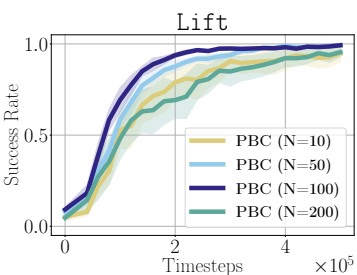

Figure 7: Comparison of DSRL finetuning performance on policies pretrained with POSTBC on Robomimic Lift, varying the ensemble size. As can be seen, POSTBC performs best with an ensemble size around 100, but is not particularly sensitive to ensemble size as long as the ensemble is not too small or too large.

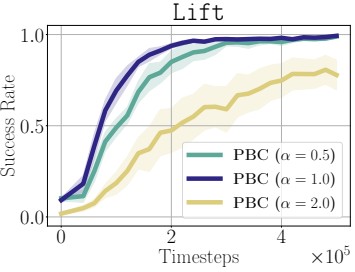

Figure 8: Comparison of DSRL finetuning performance on policies pretrained with POSTBC on Robomimic Lift, varying the noise weight $\alpha$. Increasing $\alpha$ too much typically hurts performance, and if $\alpha$ is too small performance reduces to that of BC. In general we found that setting $\alpha = 1.0$ performs well across many settings.

