# OpenReview forum: "Posterior Behavioral Cloning: Pretraining BC Policies for Efficient RL Finetuning"
_ICLR.cc/2026/Conference — Submitted to ICLR 2026_

### Official Review · Reviewer_M6JC · 2025-10-17

**Soundness:** 2
**Presentation:** 3
**Contribution:** 3
**Rating:** 4
**Confidence:** 4

**Summary:**

This paper introduces Posterior Behavioral Cloning (POSTBC), a new pretraining framework designed to make behavioral cloning (BC) policies more effective initializations for reinforcement learning (RL) finetuning. The authors theoretically show that posterior BC, which learns to clone the posterior distribution of the demonstrator’s behavior rather than its empirical action distribution, yields better action coverage without increasing suboptimality. To make this practical in continuous control settings, the paper develops an implementation using generative diffusion policies. The posterior distribution is approximated by training an ensemble of perturbed BC models to estimate posterior variance, and then fitting a policy to a mixture of BC and posterior-generated samples.

**Strengths:**

1. The paper tackles a highly relevant problem — how to design better pretraining objectives so that behavioral cloning (BC) policies serve as more effective initializations for reinforcement learning (RL) finetuning. While most prior work focuses on improving the finetuning algorithms themselves, this work shifts attention to the quality of the initialization, a perspective that is both conceptually insightful and practically important for sample-efficient RL in robotics and other domains.
2. The theoretical analysis is rigorous, well-motivated, and clearly presented.

**Weaknesses:**

1. Limited related work discussion and insufficient comparative evaluation: The paper’s experimental comparison is somewhat incomplete. While the work focuses on improving BC pretraining for RL finetuning, it does not adequately compare against other imitation learning approaches that could also serve as initializations for RL finetuning, such as DWBC, ISWBC, the DICE-family. Further, the literature review is also insufficient. As far as I know, there exists offline-to-online imitation learning, which trains a good initialization for online imitation learning.
2. Unclear treatment of the value function during RL finetuning: Since POSTBC pretrains only the policy but not a corresponding value function, it remains unclear how the subsequent RL finetuning phase initializes or learns its critic. Directly reusing a random or uninitialized value function could cause policy–value misalignment, especially in actor–critic frameworks where stable updates rely on consistent policy–value coupling. This raises the question of whether the improved finetuning performance comes purely from the better policy initialization, or if it is affected by potential instability in the critic’s early-stage learning. Furthermore, one might ask whether it is possible to derive a value function estimate from the posterior BC process itself (like a byproduct from policy learning).

**Questions:**

1. How does the performance change with the number of trajectories?
2. How does it perform with diverse quality of data?

---

> ### Author Response · Authors · 2025-11-18
> **Response to Review**
>
> We thank the reviewer for their feedback. We have run an additional imitation learning baseline, as well as an ablation on the number of trajectories (these results are included in the updated version of the paper, and we have also prepared a summary of them that can be viewed [here](https://drive.google.com/file/d/1rD4CQkARVv_g8FxIi5z6r0bWm1m4xbqw/view?usp=share_link)). We believe this addresses all concerns the reviewer raised, but if any further issues remain please let us know and we will do our best to address them as well. Please see below for additional clarifications on specific issues raised.
>
> > Limited related work discussion and insufficient comparative evaluation…
>
> We have added an additional imitation learning baseline to our Robomimic experiments, pretraining with ValueDICE [1], and find that this significantly underperforms even BC. Please see the updated version of the paper or [this](https://drive.google.com/file/d/1rD4CQkARVv_g8FxIi5z6r0bWm1m4xbqw/view?usp=share_link) document for these results.
>
> Regarding the other algorithms the reviewers suggest, we believe there are key differences between the settings where these approaches apply and our setting:
> - Methods such as DWBC and ISWBC require access to *two* offline datasets: $D_e$, the expert data, as well as $D_o$, data collected by a suboptimal policy, and critically rely on access to $D_o$. In contrast, we only assume access to $D_e$, and therefore approaches like DWBC and ISWBC cannot be run in our setting.
> - Many other existing algorithms for imitation learning require *online* access to the dynamics during training. Our focus is on simply obtaining a policy with *fully offline pretraining*, so such approaches do not directly apply in our setting.
> - The approaches we are aware of for “offline-to-online” imitation learning (such as [1]) also fall under one of the categories considered above.
>
> Given this, the majority of existing approaches for imitation learning we are aware of do not apply in our setting. However, one approach recommended by the reviewer—ValueDICE—can be run with fully offline training, and only requires access to a single expert dataset. We have added additional discussion to the related work section on other imitation learning approaches.
>
> We would like to emphasize a philosophical point on how we view our work, which we outlined in the response to all reviewers. In particular, our goal is not to propose an entirely new approach to pretraining that enables fast online improvement. Rather, our aim is to propose a simple modification to the approach already being used across machine learning—behavioral cloning. Virtually all state-of-the-art models across robotics and language rely on BC pretraining, not more sophisticated imitation learning approaches. Our approach enables a very simple modification to BC (adding a small amount of noise to the objective), which we believe is of much greater utility than more complex pretraining methods.
>
> [1] Kostrikov, Ilya, Ofir Nachum, and Jonathan Tompson. "Imitation learning via off-policy distribution matching." arXiv preprint arXiv:1912.05032 (2019).
>
> [2] Yue, Sheng, et al. "OLLIE: Imitation learning from offline pretraining to online finetuning." arXiv preprint arXiv:2405.17477 (2024).
>
> > Unclear treatment of the value function during RL finetuning…
>
> The value function learning is part of the RL finetuning method, and is independent of our pretraining approach. Our goal is to show that our approach applies for a variety of RL finetuning methods, and for each one considered, we simply use the standard, existing implementation, following the precise value learning approaches used by each of these algorithms. We have clarified this in the experiments section. Furthermore, for PostBC, BC, and $\sigma$-BC, we use identical RL finetuning methods, so the value learning is identical for each of these, and as such any difference in performance is only due to the different initialization. While improving value learning from offline data is an interesting question, this is tangential to the goal of this work.
>
> > How does the performance change with the number of trajectories?
>
> Please see our updated results [here](https://drive.google.com/file/d/1rD4CQkARVv_g8FxIi5z6r0bWm1m4xbqw/view?usp=share_link) for an ablation on the number of trajectories. We see that as the number of trajectories in the training dataset increases, the improvement of PostBC over BC decreases. This is to be expected—if we have enough data, then our uncertainty will be small, and PostBC essentially reduces to BC. PostBC therefore gracefully scales between settings where it does yield significant gains, and settings where BC is sufficient.

---

> > ### Author Response · Authors · 2025-11-18
> > **Response to Review (continued)**
> >
> > > How does it perform with diverse quality of data?
> >
> > For the Robomimic experiments, we utilize the Multi-Human Robomimic dataset, which contains demonstrations from demonstrators of three different quality levels, inducing significant diversity over the demonstration quality. Thus, our Robomimic results illustrate that PostBC can perform effectively when the quality of data differs significantly. Furthermore, Libero demonstrations are significantly less diverse and roughly of the same quality. Thus, our existing results span settings where demonstrations are from diverse quality datasets, as well as datasets of low diversity, and our results show that PostBC performs effectively across these settings.

---

> > > ### Comment · Reviewer_M6JC · 2025-11-24
> > >
> > > Thank you for the detailed rebuttal. The additional explanations indeed helped resolve part of the concerns raised in the original submission. However, after re-reading the rebuttal as well as considering the comments from the other reviewers, I still find that the overall contribution of the paper remains limited.
> > >
> > > First, the proposed approach fundamentally relies on BC, which in practice requires high-quality expert demonstrations; this assumption further limits its applicability in real-world settings where such data is difficult to obtain. Second, the authors stated that “our goal is not to propose an entirely new approach to pretraining that enables fast online improvement, but rather to propose a simple modification to the approach already being used across machine learning—behavioral cloning.” I believe that such a simple modification of BC is insufficient, as it offers limited insight into how to effectively bridge pre-training and finetuning.
> > >
> > > For these reasons, I maintain my original score.

---

### Official Review · Reviewer_dsv5 · 2025-10-31

**Soundness:** 2
**Presentation:** 1
**Contribution:** 1
**Rating:** 2
**Confidence:** 2

**Summary:**

The paper proposes posterior behavioral cloning, a pretraining approach for behavior cloning policies in order to make the finetuning processes efficient. The paper suggests a theoretical result that the standard behavior cloning often provably fails to cover the demonstrator’s distribution, while the proposed posterior behavior cloning effectively covers.

**Strengths:**

The paper has a theoretical contribution on how the proposed posterior behavior cloning could be provably better than the standard behavior cloning, in terms of the action coverage. Based on the findings, the paper proposes a simple instantiation of the posterior behavior cloning for continuous control settings.

**Weaknesses:**

While the paper explicitly states that “there do not exist any approaches which aim to pretrain policies with a BC-like objective on demonstration data, with the aim of obtaining an initialization that is an effective starting point of finetuning”, this is exactly what the typical meta-learning for supervised learning does, and behavior cloning is just an example of supervised learning (the outer loop of gradient-based meta-learning corresponds to the “posterior behavior cloning”). Even if we narrow our attention to “training policies on reward-free demonstration data”,  there still exists extensive literature about meta-imitation learning (e.g., [1, 2]) that meta-trains a policy using reward-free demonstrations by performing behavioral cloning. Therefore, I’m highly concerned about the reliability and novelty of the paper, and also about the lack of comparisons with these approaches both in theory and experiments.

[1] Finn et al., One-Shot Visual Imitation Learning via Meta-Learning, 2017.

[2] Gao et al., Transferring Hierarchical Structure with Dual Meta Imitation Learning, 2022.

**Questions:**

(continuing from the concerns in the weaknesses part) Behavior cloning is an example of supervised learning, and there exist several studies [3, 4] on the relationship between hierarchical Bayesian learning and meta-learning. Which aspects of the proposed setting (e.g., MDP, continuous control, etc.) make a difference from the typical meta-learning? Does the proposed theory directly address the difference?

[3] Grant et al., Recasting gradient-based meta-learning as hierarchical bayes, 2018.

[4] Yoon et al., Bayesian model-agnostic meta-learning, 2018.

---

> ### Author Response · Authors · 2025-11-18
> **Response to Review**
>
> We thank the reviewer for raising this comparison with meta-learning. We believe there may be a misunderstanding regarding the setting considered in this paper as compared to the meta-learning setting, however, and that meta-learning approaches do not, in general, apply in our setting. We clarify this distinction below—please let us know if this addresses your concerns.
>
> In particular, meta-imitation learning differs from our setting in three key ways:
> - Meta-imitation learning assumes access to demonstration data from *more than one task*, and attempts to learn an initialization that will allow for quickly adapting to demonstrations from a *new* task. In contrast, we are interested in the setting where we pretrain on a *single* task (though our approach does extend to multiple tasks), and aim to find an initialization that allows for improvement on the *same* task.
> - Meta-imitation learning aims to learn from *demonstrations* on the new task, while we aim to use RL to learn from (potentially suboptimal) data collected from online rollouts and labeled with rewards. While there do exist meta-RL approaches which can similarly learn from suboptimal data collected online, these require reward-labeled data in pretraining (or online access to environments), while in our case we only have offline demonstration data with no rewards.
> - A primary goal of our work is to ensure that the pretrained policy itself performs well (that is, it has effective 0-shot performance, on par with BC). Meta-imitation learning, in contrast, typically does *not* consider the 0-shot performance of the pretrained policy, but instead considers the *1-shot* performance, i.e. the performance after it has already been updated on the new task. This is a fundamentally different constraint on the pretrained policy than is typically present in the meta-learning setting, but is critical in real-world settings where we want a pretrained policy that behaves well 0-shot.
>
> We believe the meta-learning setting is fundamentally different from the setting we consider, and, given this, it is not clear how we should apply meta-learning approaches to our setting. If the reviewer could provide further guidance on how the suggested approaches apply here, that would be helpful to us in running them as baselines.
>
> We would also like to emphasize a philosophical point on how we view our approach, which we have already outlined in the response to all reviewers. In particular, our goal is not to propose an entirely new approach to pretraining that enables fast online improvement. Rather, our aim is to propose a simple modification to the approach already being used across machine learning—behavioral cloning. While it may be possible to formulate a meta-learning-like approach to obtaining an effective initialization for RL finetuning from demonstration data (though it is not obvious how to do this), virtually all state-of-the-art models across robotics and language rely on BC pretraining, not meta-learning. Furthermore, the existing meta-learning approaches we are aware of are significantly more complex than BC (for example, MAML-style approaches require computing a second-order gradient). Our approach enables a very simple modification to BC (adding a small amount of carefully selected noise to the objective), which we believe is of much greater utility than proposing an entirely novel pretraining approach.
>
> Nevertheless, we have toned down our claims in the updated version of the paper somewhat, and have also added a section to the related work highlighting how our setting differs from the meta-learning setting.

---

### Official Review · Reviewer_vh8q · 2025-11-01

**Soundness:** 2
**Presentation:** 1
**Contribution:** 3
**Rating:** 2
**Confidence:** 3

**Summary:**

The authors investigate the problem of suboptimal RL finetuning performance when using a policy initially trained via behaviour cloning. The authors provide several theoretical results showing the error bounds for achieving an optimal policy when insufficient coverage is available with the demonstration policy. An algorithm is proposed that uses stochastic labels to help prevent loss of policy coverage on actions while also maintaining initial behaviour-cloning policy performance. The authors perform experiments on Libero and Robomimic to validate their algorithmic suggestions that initial performance is maintained and that their posterior policy accounting approach enhances RL finetuning.

**Strengths:**

I like the emphasis on providing theoretical foundations to justify the model's algorithmic choices. Showing sample bounds and the effects on policy cumulative reward as a function of the number of actions, states, and timesteps provides better justification for the problem. It is more insightful than just proposing an algorithm. We like the author's solution because of its ease of implementation and potential applications in continuous control.

**Weaknesses:**

Technically, the paper has not violated the page limit, but it ends abruptly on page 9 without a clear conclusion. This ending indicates that more time is needed to condense the current draft to fit the page limit, without the additional page in the rebuttal.  Figures 2 - 5 are squished together as a byproduct of this. It isn't easy to see the content of Figure 2, and the legend in Figure 3 dominates one of the presented plots. Discussion across many sections can likely be compressed to address these issues and remove redundant discussions. Here are some ideas:
- If Algorithm 1 is not the main algorithm, it can probably be cut or moved to the appendix.
- The Equation before Equation (3) also looks quite similar, or otherwise frankly could be inline with the text. This suggestion might also apply to the Gaussian policy on lines 314/315.
- Line 309 - 311 summarizing the previous section could be cut.
- Proposition 1: Could perhaps be moved to the appendix, and the same for Mathematical Notations.

Furthermore, although we find the theoretical justifications compelling, the existing experiments at a minimum need more nuanced discussion. As one of the central claims is that BC can improve exploration during RL finetuning, it would be helpful to understand why the results in Figures 4 & 5 appear mixed in the benefits of the author's algorithm. One idea might be to modify the training data distributions to create conditions similar to those described in the theory section, or at least examine the action distributions directly to see how variance is affected across the learned policies.

Likewise,  the claims from Table 1 should be more specific. The 20% over BC and 10% success rate improvements over \sigma-BC are specific to Libero. All results are from the Best-of-N (1000 Rollouts). In other settings, we see similar performance between BC and PostBC, with \sigma-BC performing better in one case (Liberto Scene 1). The results for Libero on the 2000 Rollouts setting seem to be missing, or if there is a reason they are excluded, that should be explained.  It would also be good to conduct hypothesis tests to determine whether these results are statistically significant, thereby strengthening the author's claim.

Other Comments:
> Consider including the work on "robust policy optimization," which proposes a similar stochastic labelling approach for the online RL setting [1]. The author's project better motivates this decision theoretically, but the point is to acknowledge works that study applying a similar mechanism.
[1] Rahman, Md Masudur, and Yexiang Xue. "Robust policy optimization in deep reinforcement learning." arXiv preprint arXiv:2212.07536 (2022).

> The "Best-of-N" approach should be more clearly explained in the Background section. Several significant results rely on this, but how it is performed precisely is difficult to discern in the current paper.

> Table 1 — if this shows success rates, make that clear. The bold font is confusing, so either explain it in the caption or remove it.

> Lemma 6 in Appendix — If this is a proof for "Theorem 2," then it shouldn't be called a lemma.

**Questions:**

Definition 4.1 — What is the range of \gamma?

Proposition 2 - Is there some meaningful range implicitly accounted for in this bound? Is J(\pi^\beta) bounded between 0 and 1?

Proposition 2 - If this is an informal argument, is a proof or related citation provided to support this proposition? The proposition appears central to the paper's arguments but is not treated rigorously.

Table 1: Why are the results for Libero Best-of-N (2000 Rollouts) missing?

---

> ### Author Response · Authors · 2025-11-18
> **Response to Review**
>
> We thank the reviewer for their detailed feedback. We have revised the paper based on the suggestions provided. In addition, we have run Best-of-$N$ on Libero with 2000 rollouts, and have run several additional ablations (these results are included in the updated version of the paper, and we have also prepared a summary of them that can be viewed [here](https://drive.google.com/file/d/1rD4CQkARVv_g8FxIi5z6r0bWm1m4xbqw/view?usp=share_link)). We believe this addresses all concerns the reviewer raised, but if any further issues remain please let us know and we will do our best to address them as well. Please see below for additional clarifications on specific issues raised.
>
> > Technically, the paper has not violated the page limit…
>
> We thank the reviewer for these suggestions. Please see the updated version of the paper—we have condensed some of the discussion and incorporated many of the suggestions.
>
> > Furthermore, although we find the theoretical justifications compelling, the existing experiments at a minimum need more nuanced discussion…
>
> We have incorporated additional discussion on these points into the updated version of the paper. We would like to highlight in particular:
> - We would not necessarily expect our approach to improve in all settings. This is consistent with the theory—if the demonstration data is sufficient to determine the demonstrator’s actions already, then there is no need to expand the action distribution as PostBC does. In this case PostBC reduces to standard BC, and thus should perform similarly to standard BC. Please see our ablation on dataset size included in this document for illustration of this.
> - Our results show that there are settings where PostBC does give substantial gains over BC (see Figure 3-5 and Table 1). Furthermore, even in the cases where PostBC does not improve over standard BC, it performs no worse. Thus, PostBC allows us to gracefully shift between settings where an expanded action distribution is needed for finetuning, and settings where it is not.
> - In the period between the ICLR submission deadline and the reviews being released, we performed additional ablations on how we were computing the ensemble and found that bootstrapped sampling performs better than our original proposed approach of adding noise to estimate uncertainty. In our updated results, the difference between PostBC and BC is more substantial. Please see [this document](https://drive.google.com/file/d/1rD4CQkARVv_g8FxIi5z6r0bWm1m4xbqw/view?usp=share_link) or the updated version of the paper for these results.
>
> > Likewise, the claims from Table 1 should be more specific... The results for Libero on the 2000 Rollouts setting seem to be missing…
>
> We have added the results from Best-of-N on Libero with 2000 rollouts to the results table, and see that the performance gains of PostBC hold there as well. We note that our table already includes confidence intervals (computed using procedures standard in the RL literature, in this case the standard errors) showing that our results are statistically significant.
>
> > Consider including the work on "robust policy optimization,"...
>
> We have added a discussion on robust policy optimization to the related work in the updated version of the paper.
>
> > Definition 4.1 — What is the range of \gamma?
>
> There is no specific range for $\gamma$ in Definition 4.1–the definition is with respect to *any* chosen $\gamma$ (that is, we say a policy is a $\gamma_0$-sampler if it satisfies Definition 4.1 with $\gamma \leftarrow \gamma_0$). As our later results show, however, depending on how we clone the data we can obtain different values of $\gamma$ that will satisfy Definition 4.1.
>
> > Proposition 2 - Is there some meaningful range implicitly accounted for in this bound? Is J(\pi^\beta) bounded between 0 and 1?
>
> Yes. As stated in the prelimaries, all rewards are in [0,1], so $\mathcal{J}(\pi) \in [0,H]$ for all $\pi$.
>
> > Proposition 2 - If this is an informal argument, is a proof or related citation provided to support this proposition? The proposition appears central to the paper's arguments but is not treated rigorously.
>
> The full version of Proposition 2 is stated as Proposition 5 in the appendix (with full proof). We stated the informal version in the main text as the formal version is somewhat technical, and we felt that this detracted from the flow of the paper. We have added a more explicit reference to the formal version in the main text to avoid confusion over this in the future.

---

### Official Review · Reviewer_119R · 2025-11-01

**Soundness:** 3
**Presentation:** 3
**Contribution:** 3
**Rating:** 4
**Confidence:** 3

**Summary:**

This paper proposes **Posterior Behavioral Cloning (POSTBC)**, a pretraining method that aims to make behavioral cloning (BC) policies more suitable for reinforcement learning (RL) finetuning. Rather than directly imitating demonstrations as in standard BC, POSTBC models the **posterior distribution of the demonstrator’s actions**, enabling higher entropy (exploration) in uncertain regions and more deterministic behavior in data-rich states. The authors provide theoretical results showing that standard BC can fail to cover the demonstrator’s action space, while POSTBC ensures coverage without compromising imitation performance. Using diffusion policies, the method demonstrates improved **sample efficiency and finetuning effectiveness** on robotic control benchmarks (Robomimic, Libero), achieving consistent gains over standard BC without degrading pretrained policy performance.

**Strengths:**

- The idea of leveraging a posterior distribution over actions for better RL finetuning efficiency is conceptually novel and practically meaningful.
- Clearly identifies and addresses a core limitation of standard BC in the context of RL finetuning.
- Provides a well-developed theoretical framework with sound reasoning about action coverage and RL-finetuning potential.

**Weaknesses:**

- If I understand correctly, the authors perturb the training actions with Gaussian noise, train multiple policies on these perturbed datasets, and compute the covariance of their predicted actions to approximate a posterior distribution. This approach seems somewhat ad hoc. If the base models have sufficient capacity to memorize the data, the estimated covariance would simply reflect the injected noise, effectively collapsing back to the σ-BC baseline. In classical bootstrap ensemble methods, one typically resamples data subsets rather than adding noise—this would arguably provide a more principled estimate of epistemic uncertainty.
- The experimental evaluation, while supportive, feels limited in scope and lacks deeper analysis or ablation (e.g., number of demonstrations, impact of ensemble size, or posterior weight α).

**Questions:**

1. The Libero benchmark does not officially provide reward functions. How were rewards defined in your RL finetuning setup? Did you use binary success/failure rewards or custom rewards?
2. Line 400 mentions limiting the number of demonstrations. Please clarify the exact number used per task and include an ablation study on this parameter. Intuitively, the gain of PostBC should decrease as the number of demo increases.
3. Have you considered using *bootstrapped resampling* (i.e., different subsets of the demonstration data) instead of Gaussian perturbations for ensemble training? This might yield a more faithful posterior estimate.

Overall I like the problems studied by this paper, I'd love to adjust my rating if the authors can settle my concerns.

---

> ### Author Response · Authors · 2025-11-18
> **Response to Review**
>
> We thank the reviewer for their detailed feedback. We have run the requested ablations, and have updated results using bootstrapped sampling to compute the ensemble (these results are included in the updated version of the paper, and we have also prepared a summary of them that can be viewed [here](https://drive.google.com/file/d/1rD4CQkARVv_g8FxIi5z6r0bWm1m4xbqw/view?usp=share_link)). We believe this addresses all concerns the reviewer raised, but if any further issues remain please let us know and we will do our best to address them as well. Please see below for clarifications on specific issues raised.
>
> > If I understand correctly, the authors perturb the training actions with Gaussian noise…  This approach seems somewhat ad hoc…
>
> We would like to highlight that our approach to quantifying uncertainty is not novel to this work, and has been well-studied in both the theoretical (see e.g. [1]-[3] for a formal justification of this method) and empirical [4] literature. Furthermore, as Proposition 4 shows, adding random noise to the dataset *does* allow us to measure uncertainty, and this is independent of model capacity.
>
> As an example, assume $\mathfrak{D} = [(s\_1, a\_1), (s\_2, a\_2), … , (s\_2, a\_{N+1})]$, where $s_2$ is observed N times, and each noisy member of the ensemble is $\mathfrak{D}\_i = \{ (s_t, a_t + w_t^i) \}_{t=1}^{N+1}$ for $w_t^i \sim \mathcal{N}(0,1)$. Then, even if we have full model capacity, ensemble $i$ will predict $a\_1 + w\_1^i$ at $s_1$, but $\bar{a}\_2 + \frac{1}{N} \sum\_{t=2}^{N+1} w\_t^i$ at $s_2$, for $\bar{a}\_2 = \frac{1}{N} \sum\_{t=2}^{N+1} a_t$. At $s_1$, each member of the ensemble has prediction distributed as $\mathcal{N}(a_1, 1)$, so the ensemble’s variance will be 1 at $s\_1$. However, at $s\_2$, each member of the ensemble has prediction distributed as $\mathcal{N}(\bar{a}\_2, \frac{1}{N})$, so the ensemble’s variance will be $1/N$ at $s\_2$. We see that this is consistent with the amount of uncertainty we have in $D$ for each point $s\_1$ and $s\_2$, and that this holds regardless of model capacity.
>
> [1] Russo, Daniel. "Worst-case regret bounds for exploration via randomized value functions." Advances in neural information processing systems 32 (2019).
>
> [2] Qin, Chao, et al. "An analysis of ensemble sampling." Advances in Neural Information Processing Systems 35 (2022): 21602-21614.
>
> [3] Kveton, Branislav, et al. "Randomized exploration in generalized linear bandits." International Conference on Artificial Intelligence and Statistics. PMLR, 2020.
>
> [4] Osband, Ian, John Aslanides, and Albin Cassirer. "Randomized prior functions for deep reinforcement learning." Advances in neural information processing systems 31 (2018).
>
> > Have you considered using bootstrapped resampling…
>
> We did in fact experiment with this approach following the ICLR submission and before the reviews were released, and found that it did indeed improve performance. Please see the document [here](https://drive.google.com/file/d/1rD4CQkARVv_g8FxIi5z6r0bWm1m4xbqw/view?usp=share_link) and updated paper for these results.
>
> > The experimental evaluation, while supportive, feels limited in scope and lacks deeper analysis or ablation (e.g., number of demonstrations, impact of ensemble size, or posterior weight α).
>
> Please see the document [here](https://drive.google.com/file/d/1rD4CQkARVv_g8FxIi5z6r0bWm1m4xbqw/view?usp=share_link), as well as the updated version of the paper, for ablations of all the factors you suggest. We believe this addresses the concerns the reviewer has raised on the limited scope of the experiments, but please let us know if any issues remain.
>
> > The Libero benchmark does not officially provide reward functions…
>
> For both the Libero and Robomimic experiments, we used a 0/1 success reward, in either case using the built-in success detector for the environment to determine this reward. We have clarified this in the experimental details in the updated version of the paper.
>
> > Line 400 mentions limiting the number of demonstrations…
>
> Please see Tables 5, 8, and 9 for the exact number of demonstrations we used in each experiment for Robomimic (and, as stated in Section C.2, for Libero we take 25 demos for each task). As noted, we have added an ablation to highlight the dependence on the number of demos, and observe that as the number of trajectories in the training dataset increases, the improvement of PostBC over BC decreases. This is to be expected—if we have enough data, then our uncertainty will be small, and PostBC essentially reduces to BC. PostBC therefore gracefully scales between settings where it does yield significant gains, and settings where BC is sufficient.

---

### Author Response · Authors · 2025-11-18
**Response to All Reviewers**

We thank the reviewers for their detailed feedback. We have updated the paper to incorporate many of the suggestions, and have also run additional experiments (including **additional requested ablations, baselines, and improved results on Robomimic**), with the results summarized in [this document](https://drive.google.com/file/d/1rD4CQkARVv_g8FxIi5z6r0bWm1m4xbqw/view?usp=share_link ) and incorporated in the updated version of the paper (all updates in blue). We would like to highlight several key points:
- **Theoretical contributions**: All reviewers acknowledge the theoretical contributions of our work, but we would like to emphasize the significance of these results. These are novel results on the fundamental behavior of RL finetuning on BC-trained policies—the workhorse algorithm in much of AI currently. They illustrate a key shortcoming of BC, and propose a simple approach that provably addresses this shortcoming. We believe these results are a significant contribution on their own, apart from our experiments, and that the experimental results further validate their significance. We would therefore ask the reviewers to re-consider the overall assessment of the paper in light of this, and evaluate our work just as much based on its theoretical contributions as our experimental results.
- **Improved results on Robomimic**: Our approach, PostBC, first computes approximate posterior samples, and then fits a policy to these posterior samples. While the original submission presents a particular method to generate approximate posterior samples—fitting predictors to “noisy” versions of the dataset—PostBC is agnostic to the precise way these samples are generated. Following the ICLR submission deadline, we continued to experiment with other ways of generating these posterior samples, and found that bootstrapped sampling—where we fit a predictor to a dataset instead generated by sampling from the original dataset with replacement—leads to improved performance, allowing PostBC to outperform existing approaches by a larger margin on Robomimic (in fact, the original Libero results utilized bootstrapped sampling, as was noted in our experimental details section). We have updated the paper to reflect these results.
- **Motivation for the paper and choice of baselines**: We would like to highlight a philosophical point on how we view our work. Our goal in this work is not to propose an entirely new approach to pretraining. Rather, our aim is to propose a simple modification to the approach already being used across machine learning—behavioral cloning. While it may be possible to formulate a variety of more sophisticated pretraining approaches, virtually all state-of-the-art models across robotics and language rely on BC pretraining. Our approach enables a very simple modification to BC (adding a small amount of carefully tuned noise to the objective), and can easily be integrated into any BC training pipeline, which we believe is of much greater utility than proposing an entirely novel or more sophisticated pretraining approach. Our discussion and baselines were chosen around this viewpoint—our aim is not to compare against *every possible method for training from offline demonstration data* but instead to show that *our approach is significantly better than the approach that is by far the most widely used approach in practice*.

---

### Meta-Review · Area_Chair_jKAA · 2026-01-07

**Summary:**

This paper proposes Posterior Behavioral Cloning (POSTBC), a pretraining method that aims to make behavioral cloning (BC) policies more suitable for RL finetuning. POSTBC models the posterior distribution of the demonstrator’s actions, enabling higher entropy (exploration) in uncertain regions and more deterministic behavior in data-rich states.

Reviewers were primarily concerned with the technical implementation of uncertainty estimation (noise vs. bootstrapping), lacking proper theoretical justifications, poor experimental comparisons with baselines - insufficient experimental comparison against meta-learning and imitation learning baselines (e.g., ValueDICE), and presentation issues. There was also a shared skepticism regarding the magnitude of contribution, with some reviewers questioning whether a "simple modification" to behavioral cloning (BC) constituted a sufficiently novel advancement for the conference.

**Reviewer Concerns:**

The authors addressed some technical critiques by implementing bootstrapped sampling, providing missing 2000-rollout data for Libero, and adding the requested ValueDICE baseline and a conclusion. However, the novelty and conceptual framing remain outstanding points that have not been addressed.

**Reviewer Scores:**

All reviewers would probably maintain their score since the main issue of limited technical novelty, and insufficient comparisons with baselines still exists. All reviewers are leaning negative.

---

### Decision · Program_Chairs · 2026-01-26

Reject